# Family-Centered Design: Interactive Performance Testing and User Interface Evaluation of the Slovenian eDavki Public Tax Portal

**DOI:** 10.3390/s21155161

**Published:** 2021-07-30

**Authors:** Jure Trilar, Tjaša Sobočan, Emilija Stojmenova Duh

**Affiliations:** ICT Department, Faculty of Electrical Engineering, University of Ljubljana, 1000 Ljubljana, Slovenia; Tjasa.Sobocan@ltfe.org (T.S.); emilija.stojmenova@fe.uni-lj.si (E.S.D.)

**Keywords:** family-centered design, government digital services, user interface, web accessibility, user performance, user evaluation, unmoderated remote testing

## Abstract

This is the latest article in a series of research on the family-centered design concept. The theoretical context was revisited and expounded to support its usefulness in conjunction with a user-centered design approach within distinct application domains. A very important contribution is applied through the development of the instruments—website capture, a public testing platform, results processing and the Web Content Accessibility Guide 2.1 recommendation tool—to conduct unmoderated remote testing of user interfaces that corresponds to the requirements of general digitalization efforts as well as the response to current and future health risks. With this set of instruments, an experiment was conducted to address the differences in usage, and performance-wise and user-based evaluation methods, of the eDavki public tax portal, among two generations, adults and elderly citizens, and between an original and an adapted user interface that respects accessibility and other recommendations. The differences found are further discussed and are congruent to particularities that have been modified within interfaces.

## 1. Introduction

In the modern age, the family is not constrained to household organization or other established models; rather, it can also function as a multigenerational nano social network that maintains many functions leveraged with the use of digital technology. Presently, it is not uncommon that all generations of family members interact with digital solutions on a daily basis, connecting among themselves, their interests and different kinds of private and public digital services. The use of digital technology is not limited to younger or young adult generations, as there is a growing demand for the elderly to interact with these information communication systems in some capacity in order to stay as autonomous as possible in their own home and environment. In the current period of restricted mobility due to Covid-19, the use of digital technology for different types of support for the elderly is becoming even more evident [1]. Citing W.H. Thomas from *What are old people for*: “The most elder-rich period of human history is upon us. How we regard and make use of this windfall of elders will define the world in which we live.” [2]

In previous research work, Trilar et al. [3,4,5] focused predominantly on how to develop a different approach and prototype new digital solutions that connect family members to each other, enable communication, and promote quality, family-oriented, leisure, and an active, health-friendly lifestyle between generations with the help of sensors and information communication technologies (ICT) in the wider context of a smart city. A concept for appropriate digital product and solution features and presentation design that facilitates the type of planning, considered as a part of a broader research work—a human-centered digital transformation [6]—was proposed.

Through their work so far, Trilar et al. tried to establish a family-centered design (FCD) concept for creating modern digital solutions. Based on the results of preliminary exploration, interactive prototypes based on scenarios that foster inclusion and benefit all generations involved were developed. For further development of FCD, in the current part of the research work, it would be desirable to use this family-friendly approach to improve existing solutions where individuals in everyday life interact with important government institutions, such as tax authorities. eDavki, the Slovenian public tax portal, has been chosen due to many potential benefits for citizens and its expansive usage in the arena of government digital services. An essential group in this study was the elderly, as the authors perceived the need for improved access and usability of digital services for groups with lower digital literacy as being profoundly important. To the extent that the results could indicate that these improvements are possible, not only when designing new digital solutions, but also in the enhancement of existing ones, they would be a good indication of the usefulness of the FCD approach. Thus, experiment objectives addressed in this article include primary comparative testing of user performance and subjective user evaluation of the original platform interface and an adapted FCD-based interface version of the eDavki tax portal among generations. To support the experiment’s objectives, a proprietary testing platform was designed to correspond to the need of performing remote usability tests with the aim of stimulating users to participate through a simpler and more comfortable testing process, the evaluation of which was an additional goal during the experiment.

Amid the Covid-19 pandemic, which started in Slovenia in the second quarter of 2020, serious limitations on how to carry out the already planned user interface testing activities, which, so far, have usually taken place in the laboratory environment, became apparent. Alongside the measures to prevent the spread of the virus, the realization came that direct contact with external test participants would not be possible in the near or mid-term future due to changed social habits and recurring waves of infections. In addition, direct contact with certain demographic groups that are of particular interest to us, such as the elderly, will probably not be possible for a longer period of time.

The innovation, in comparison to other remote testing tools, was materialized in new possibilities of an automated remote testing process that can adhere to specific design approaches, in this exemplification, family-centered design, which focuses on the generational traits of the users and customized interfaces. The testing platform offers, in addition to the automated processing of the measured responses (time on task, success rate, number of clicks), user experience (UEQ questionnaire), usability (SUS questionnaire) and task work-load assessment (NASA-TLX questionnaire), also a novel mechanism for analyzing to what extent certain criteria within the W3C Web Content Accessibility Guidelines (WCAG) 2.1 [7] and relevant instructions are respected in each visual interface while generating recommendations for their improvements.

The structure of this article, after the introduction, involves reconsideration of user-centered design methodology and the family-centered design approach in the Background section, followed by the presentation of the testing platform and interfaces in the Instruments section. After that, the Experiment section expounds on the method used, the sample of participants, and variables that are pertinent to the experiment, before analyzing its results. Discussion of the results accompanied with Implications and limitations of the experiment and tools used is followed by Conclusions to summarize the research efforts.

## 2. Background

In this section, the existing methodology of user-centered design (UCD) was examined, along with the potential of the FCD concept in terms of the intergenerational and multi-family life ICT use characteristics, in order to define how to best structure and address user requirements and needs in connection with the establishment of an interface infrastructure that would respond to the needs of the target group. This section entails a literature overview and analysis that serve as a foundation to the experiment and prototype development. The relevant aspects of the UCD approach, usability and user experience aspects (UX) were integrated while remaining applicable to the concept of FCD, and used within our experiment.

### 2.1. User-Centered Design

UCD focuses on building an effective design of a user interface and processes for digital artifacts, therefore aiming to improve the quality of interaction between the user and the product [8,9]. The foundation of the UCD approach refers to the process applied when engineering an experience, with the goal of putting the user at the heart of the design process, and thereby enabling an understanding of the wants, needs and limitations a user could face. It should entail software design with an understanding of the user, which includes the user in every step of the design, and it should allow iterations with user’s feedback incorporated [10]. According to Pratt and Nunes [11], “understanding who we design for, what they want and need, and the environment in which they will use our designs, is not only a good way to guarantee a successful product, but also a safer, saner world”. This way, the final outcome is an optimized and user-friendly product that holds higher chances for faster adoption by the users.

The UCD approach changes due to context, technological development or even as a result of the goals of the design, especially as continual testing with users is an essential part of UCD. In a systematic approach, a UCD process that includes five phases is used: first, a design strategy to integrate business objectives with a new solution and define the relationship between the two; next is user research, to identify and define design goals, followed by concept design and production of a prototype, after which the usability testing is implemented, with evaluation as the last step [12,13]. To create intuitive and user-friendly digital services and products, UCD considers and applies the following design characteristics: information architecture, ergonomics, market acceptance, user preference, functionality/utility, coding, and aesthetics, which cover the design process from start to finish [14].

The task of an engineer is to revise the prototype based on user testing iterations. This way, the final result should comply with users’ needs, situations, limitations, behavior, etc. Nevertheless, the UCD process does not end here; rather, it continues to strive toward additional improvements, or create new products based on the UX. When designing the interface to test our FCD concept, it was necessary to consider the living characteristics and circumstances of the ecosystem where multi-family systems operate. This was achieved by defining personas and their needs within the FCD concept, considering that multi-family systems are living, constantly changing environments (for example, the needs of the elderly change faster with aging and require higher adjustment efforts for the whole multi-family system in comparison to other social groups). The way people interact with technology is evolving rapidly, which also changes relations between society and (public) digital services that people need for functional living. This again shows that UCD is a continual approach that grows with the product development and user involvement in order to improve and scale UX [14,15].

It is important to include the user in the design process; however, the end results should not rely solely on the users’ feedback in all cases. When creating something innovative, especially in a digital sphere that is yet unknown to the existing end users, we should take into consideration that our product can also create a new market and lead to radical innovation [16].

### 2.2. User-Centered Design Methodology and Standards

User-centered design is part of the ISO standards, namely ISO 9241-210: 2019 Ergonomics of human-system interactions in the part that addresses user-friendly features of interactive systems [17]. The definition of system usability is an integral part of IEEE Std. 61012, 1990, which defines it as the ease of use, learnability and ability to recognize the results, the output of a system or component [8]. ISO/IEC 9126-1: 2000 describes it as a feature that makes the software or product understandable, usable, learnable and attractive when used under established conditions [8].

Jacob Nielsen, one of the most established experts in the field of usability testing of human–computer interaction on online and mobile user interfaces, defined five qualities in usability goals: learnability (easy to learn), efficiency of use (no unnecessary resources for the user), memorability, few and non-catastrophic errors (used without causing errors), and satisfaction (using the interface should be pleasant) [18]. Additionally, as usability is close to acceptance, Davis [19] clarifies acceptance as the user’s intention to use the product, depending on their perceived usefulness and perceived ease of use. Although technological and design approaches toward creating efficient systems have advanced, these essential keystones remain. These pillars were supported throughout the testing of the FCD concept to objectively determine whether this approach is usable within the broader UCD framework.

Nielsen also defined a usability problem [20] as “any aspect of the user interface which might create difficulties for the user with respect to an important usability indicator (such as: ease of understanding and learning how to operate, ease of use, time to complete the task, or subjective user satisfaction)”.

According to Gulliksen et al. [21], UCD is represented in three principles, while ISO in its updated version accounts for six principles:The design is based upon an explicit understanding of users, tasks and environments.Users are involved throughout the design and development process, meaning there is an early focus on users regarding the functionality of a product. The higher the intended usability of the product, the greater the extent that users must be involved in its design. In this way, fewer corrections need to be made at later stages in the software development. It is essential to identify the users’ needs, preferences and the current manner of solving the actual problems related to system functionality.The design is driven and refined by user-centered evaluation: empirical measurement of product use. In this principle, the main emphasis is on facilitating tests and effectively documenting mistakes through various metrics: measuring errors, completing scenarios, time spent on exploiting the system’s functionalities, subjective user experience responses, usability, etc. While testing, users typically test a prototype or end-product by completing distinct scenarios that include the main functionalities of the product.The process is iterative (iterative development), which recommends that all stages of development are implemented repeatedly until satisfactory results are achieved.The design addresses the whole user experience.The design team combines multidisciplinary skills and perspectives.

In the experiment of FCD testing, all six principles were considered, with a particular inclusion of users early on, to achieve the most viable and functional product. Identifying users’ needs, and understanding how they currently face and solve problems on the topic of our design, was essential for creating a solution-based digital product (interface).

There are various lists of UX design principles that usability experts could use. In 1994, Nielsen presented the 10 heuristics principles on usability that are most applicable when creating a web interface or a digital product. These 10 principles serve as broad guidelines or a framework, as not all principles are applicable to all evaluation scenarios, so there is a common practice that UX designers adopt between five to ten Nielsen’s principles and/or add their own. As technology is changing rapidly and constantly, design engineers and experts are adding user-centered principles that emphasize respecting the users, their capabilities and privacy, as well as them having a pleasant experience with the product. Although Nielsen’s principles originate from the 1990s, they are still largely used or complemented by other principles (for example, following the task progress/system status, using language understandable to users, error prevention, etc.) [22]. This shows that UX testing is a living, evolving discipline that adapts to the needs of the users. For example, users prefer to see the progress they are making if an interaction is required from them on the website, so this should be regarded under the principle of visibility of the system status [23]. 

To accommodate evolving approaches, terminals, visual culture and user engagement within the field of UX, there is a necessity to allow for common quantitative information that would provide some comparison. Established metrics and questionnaires in performance- and questionnaire-based evaluation user interface testing [24] were used in the research team’s previous efforts [4] in designing a new progressive web application, MyFamily, which incorporated original task scenarios that were tested in a laboratory environment and produced results that confirmed the applicability of the FCD approach. The methods for user interface evaluation used in this experiment are identical to previously used ones, with the exception that an online, remote testing environment is used for the testing of established public digital services instead of a laboratory environment, and conformance to WCAG is also included.

Standardized metrics used within the experiment consisted of two types:Performance metrics measured through objective categories, the time used and task completion:oEffectiveness indicates the probability of completing a task. A performance indicator is a favorable result of a task that a user accomplishes in a given system [24,25]. In doing so, these are measures of:▪Task completion. If a user has completed the actions required by the instructions, it is marked accordingly.▪Task errors are marked accordingly and relate to the user not being able to complete the actions required. Task errors provide insight into the most critical points when the user operated the system.oEfficiency indicates the amount of effort to complete a task. Task completion time appears in the literature [24] as the primary indicator of effectiveness. According to Frøkjær [25], the efficiency of solving a problem can also be a combination of the percentage of successful solutions and the time taken to solve the problem.User evaluation as a subjective response of the users to the system usage:oSystem Usability Scale (SUS) for measuring the usability of systems [26]. The SUS Questionnaire is a simple and familiar tool for assessing usability, which is widely used in the context of digital systems. It includes 10 items on which the respondents indicate their agreement with the five-point Likert scale: from “I disagree completely” to “I completely agree”. The SUS includes indications of learnability (items 4 and 10) and usability (other items) [27]. It also encompasses effectiveness, such as the ability of the user to complete a task using the system; efficiency, as a measure of the resource usage (time and attention) in performing a task; and satisfaction, as the user’s subjective response to the use of the system [5].oUser Experience Questionnaire (UEQ) for measuring user experience [28]. The User Experience Questionnaire contains six components, with a total of 26 items, marked with a 7-point Likert scale. It measures attractiveness as a general impression of a product; perspicuity, as product recognition; efficiency, as the ability to complete tasks without undue effort; dependability, as a measure of user control over the interaction; stimulation, in conjunction with excitement and motivation to use the product; and novelty if the product is innovative and creatively designed. Perspicuity, efficiency and dependability display pragmatic qualities and are goal-oriented, while stimulation and novelty are hedonic aspects. Attractiveness is a combined trait [29].oNASA-Task Load Index (NASA TLX) for measuring subjective load rating [30]. The subjective workload questionnaire consists of six factors: mental demand, which measures how much mental and perceived activity was required; physical demand, measuring how much physical effort was required; temporal demand, assessing how much time pressure a participant feels when completing tasks; own performance contains an assessment of one’s own performance; effort, an overall assessment of how difficult the task was; and frustration, assessing how much non-safety, irritation, stress and disruption a participant experienced to complete the task. The three factors, mental, physical and temporal complexity, relate to the participant, whilst the auxiliary three, effort, frustration, and performance, relate to the task. Each component is represented in a rank scale of 0.100 [30].

### 2.3. Family-Centered Design Concept

An upgrade of UCD with an approach that additionally considers the dynamics of heterogeneous, close-knit groups in terms of demographics and digital competencies, such as families, was proposed. This is why it was defined as a family-centered design (FCD) concept.

The intergenerational family is defined as two or more generations who may live in the same or different locations and provide a mutual support system, which is often based on reciprocity [31,32]. This also accounts for a continuous need for communication, information sharing and family engagement [31]. As each generation has its own characteristics, needs and requirements, the interaction varies from care-related issues to health, and family well-being. In this regard, ICT and digital tools offer extended support to existing infrastructure, while being able to provide most real-time information for all parties included.

The general characteristics of family systems include being heterogeneous, subject to constant changes (demographic, geographical distance), and possessing various levels of skill sets, especially digital skill sets. When it comes to technology and digital competencies, there is a large digital gap between generations, which depends on various factors, such as ICT infrastructure (computers, smart phones, tablet computers, smart wearables, etc.), access to the internet and motivation for learning digital skills. As has been explored in previous research, ICT is blurring the boundaries between work–life arrangements, which affects social interactions, and family life organization (fewer in-person interactions, increased digital gap, redefined roles, etc.). On the other hand, it can also connect families, improving their dynamics and increasing feelings of safety and belonging [32].

When looking at Slovenia’s progress in digitalization, drastic advancement in functional digital literacy cannot be foreseen. The DESI index for 2020 ranks Slovenia in the 16th place, just below the EU average, where the country lags behind in using the web and gaining new ICT skills in comparison to other EU countries [33]. Access to the internet is available in 90% of all households with at least one household member aged between 16 and 74 years. Despite that, the national research by the Statistical Office of the Republic of Slovenia, conducted in 2017, shows there are still 22% of people in the aforementioned age group without digital skills, while 57% of seniors aged between 66 and 74 years have never used the internet [34]. Although there was a major increase in digital users due to Covid-19, as online presence was necessary for a continuous functional life (distance learning and teleworking), it does not necessarily bring higher involvement of the elderly as ICT users.

ICT tools and online participation are necessary as they give access to relevant information, not only for everyday living, but also for managing health-related issues and accessing digital public services. With that, ICT can increase the chances of an active inclusive living, enabling functional living, especially of older adults [35]. Different generations have different motives to function online. With seniors, the motivation is to have ongoing communication with their grandchildren, while adults are striving for extended care and health management of their senior parents [32].

However, web usability and UX differ with age. In the two categories of older adults and general adults, there are significant differences in using ICT due to the different levels of capabilities and shifts in physiological functions. The main differences are in the decrease in cognitive activity with aging, such as concentration issues, slower memory and processing of information, and decreases in physical activity, such as declining sight, trouble using fast-moving commands (touch systems and interactive screens), and the decline of other motor skills, the loss of skin elasticity (relevant for using fingerprint technology), etc. [35]. With aging, older adults experience more difficulties in learning how to use new digital products, so the demand for qualitative usability is higher.

On the other hand, older adults have longer attention spans than younger people, and are more patient in comparison. As Chiu [35] points out, “a digital interface design for the elderly population is expectedly more complex than such a design for younger users, making the optimum interface design more difficult to achieve using standard user-centered design techniques”.

When designing (digital) products using the FCD concept in order to achieve its meaningful use-case scenarios, diverse factors of long-lasting communities, such as important life events, or goals and constraints within families, should be considered according to Trilar et al. [4]. The family-centered approach covers the use of information and communication technologies in the family, reflections on common modes of communication, assessment of access to modern technologies and digital competencies in different generations of family members, and care being taken to ensure that processes are as inclusive as possible in order to create a better (user) experience for all family members. Through important life-event-associated tasks, common goals and goals concerning the developmental tasks of different generations [36], or the stages of psychosocial development [37], are reconciled.

There were several characteristics that illustrate key features and objectives of the FCD concept as an upgraded branch of UCD [3,4] presented:In the FCD approach, the users are interconnected and dependent (families with long-lasting daily relations), and share a living environment (joint communities).Multi-user participation and, therefore, inclusive processes should be considered for a better UX (involving experiences and the nature of interactions—emotional transactions).User scenarios should also be multi-user oriented (interactions between family members have a long-term influence on the individual user’s life, interactions have a shared purpose on a daily basis).Participation of multi-user groups in the FCD approach is inclusive in all phases of design, implementation, iteration and improvements.Task performance is more complex (users—family members—can provide complementary support with ICT due to variation in digital skill levels within groups).There is a need to search for optimal and common ICT modes of communication to satisfy multi-user groups.

In this regard, the main point of differentiation between UCD and the FCD approach that could be tested within the implemented experiment lies in two aspects: first, the FCD approach considers various levels of digital skills between generations, and second, it considers inclusive processes for a variety of generations for better UX. In the following sections, this key differentiation is supported in the implemented experiment.

The goal is to measure if the distinct approach can provide better usability in terms of effectiveness, efficiency and satisfaction. Learnability and memorability, usually revealed in longitudinal and multiple recurrence testing, will not be directly observed as “a learnable system is not always efficient” [38]. Rather, the aim is to design an intelligible and coherent interface that would support those principles and qualities.

## 3. Instruments

In order to test the usefulness of the FCD approach in existing digital solutions that are relevant and meant to be used by several generations, it was decided that the usage scenarios encountered by adult citizens would be tested in different interfaces in eDavki—the Slovenian government tax portal. The portal was chosen due to the growing role it has in the everyday life of Slovenians. Although the percentage of eGovernment services in EU28 is relatively low compared to leading members [39], there is significant space for the advancement of eGovernment services delivery and information access in Slovenia. The eDavki portal is significant in digital interactions of individuals with the public officials in Slovenia, rising from 15.8% in 2019 to 25.2% in 2020 [40]. Together with other government portals, it represents the forefront of the digitalization of the public administration. Even without concern for the currently ongoing health risk crisis that has resulted in a great nudge in digital transformation activity for the whole society, public administration is one of the sectors that has great potential to benefit from the technological advances in big data management, possibilities of access through different terminals, process automation and other information-related technologies—especially from the end-users’ point of view, where the possibility of interacting with public services in a remote mode includes significant time savings and a more structured, secure and data-enhanced process. Digital public services in this context have enormous potential due to continuous accessibility. Especially during a pandemic, citizens save a considerable amount of time when interacting with public administration representatives, not to mention the automation capacity these technologies offer in terms of effortlessly handling the majority of typical cases and forms.

eDavki is an eGovernment portal and enables users (i.e., taxpayers) to digitally interact with the Financial Administration of the Republic of Slovenia (FURS) through two-way communication, e.g., submitting and receiving documents [41], as represented in Figure 1a. The portal was introduced in 2003, and until 2018 there were no significant upgrades in terms of user experience or terminal access options. In 2018, new additions in the form of a mechanism for electronic document serving, access without the government-issued certificates for natural persons, SMS notification, an UPN payment order module, a dedicated mobile application and also some user interface upgrades were implemented. The user interface was enlivened through roughly following the overall decree and annexes on the Government’s and State Administration’s integrated graphic image [42]. In retrospect, it was speculated that the implementation did not completely follow an in-depth and comprehensive design strategy or system, e.g., as in the case of the United Kingdom government portals [43]; rather, it was a product of the assessment of the user needs and technical possibilities of the contractor responsible for the implementation. The user experience was enhanced, in addition to an overall simplification of the structure, to serve the needed information in a faster manner. Optional accessibility enhancements can also be accessed through settings. The mobile, responsive interface was implemented in the public part of the portal, and the private section introduced a unified repository for all tax forms, yet without a dedicated mobile variant of the layout. 

With the use of the captured eDavki interface, which emulates the original interface as consistently as possible, and adapted interface variants with modifications in visual representation that respect relatively new web content accessibility principles, and guidelines that correspond to the FCD approach, the experiment was conducted within a testing platform that allowed for automated result processing of measured and subjective test-user responses as well as user interface features, mining and comparing them to objective and established criteria, and thus, providing computed recommendations. After considering the transferability and ease of implementing it for the most accessible and popular web server hosting configuration, the platform was built upon a widely used PHP, HTML5 and JavaScript web programming stack with dynamic calls to a MySQL database. In the following sections, the individual components that are comprised in the testing platform and are relevant for the experiment are described.

### 3.1. Task Scenarios

The task scenarios are enactments of certain type of work that the participants perform using the tested product. Typically, a task scenario would be constituted of several tasks to achieve a target action [44]. A pertinent task scenario was conceived to enable a comparative analysis in discrete interfaces. Two different interface design and implementation approaches were applied: existing techniques and methods, and implementation based on the recommendations of the UCD and family-centered design approach. Differences between them were compared using established metrics in performance-based and questionnaire-based evaluation user interface testing [24], a set identical to that used in previous research of Trilar et al. in the FCD concept implemented in the MyFamily application prototype [3]. The experiment excludes minors, as it focuses on the scenario-specific use of digital services among young adults and the elderly. 

When designing the task scenario, it was assumed that users should not be familiar with the process, as in the case of a well-known income tax correction online form. The improvements were envisioned in site locations that would not be subject to regular use and are rarely visited by users, e.g., once a year or less. This was proposed to minimize the learning and familiarity effect, thus promoting a greater need for an even more intuitive use within a relatively complex task scenario comprised of simpler tasks. 

The tasks in a concrete testing task scenario followed each other linearly in each interface, and included:Locating and logging into the private section of the eDavki portal.Locating a form for tax payment in a maximum of three monthly installments for natural persons.Completing and submitting this form to a tax authority.

Among researchers, there was the awareness present that the selected scenarios could only partially address the testing of the concept of family-oriented development and related components, yet it was imperative to test for a clear distinction between the original and the FCD-improved eDavki portal. The assumption that overreaching complexity in terms of terminals (desktop or mobile), user interfaces (time-based media, etc.), or scenarios (for different societal groups) would not produce usable, interpretive and comprehensive results due to the involvement of too many factors was assessed.

### 3.2. Interfaces

A distinct experimental design, similar to this method, to incorporate adaptive user interfaces with an approach congruent with human-centered design principles was proposed by some authors [45]. This approach entailed modulating functionalities and visual representation through 23 high- and low-level user characteristics. It was assessed that, for practical reasons, it was not required to design complex, but nevertheless very comprehensive, integral user models—this was possible in previous research when designing a dedicated family progressive web application (PWA), hence the focus on elementary user traits related to demography (and role in the family); thus, establishing fewer factors was essential. 

In the development of the FCD concept, the generational attribute was intimately connected to developmental goals (a term derived from psychology) that are frequently related to the individual’s role in the family. With this, an adaptive user model based on stereotypes that relies on information regarding the target user demographics, which are retrieved from a demographic questionnaire at the beginning of the testing process, was constructed.

The users were classified in two groups, adults and the elderly, based on their attribution to a specific set of developmental goals described in previous work of Trilar et al. [4]. Testing participants could select mixed options from both sets in the questionnaire, and were categorized in terms of their prevalent affinity to adult or elderly developmental tasks. To control for possible difficulties in this approach, there was additional input of the year of birth, enabling of post-testing inquiry in the congruence of classification, and assignment of generation according to the age variable. In this case, the adults were 18–64 years old, while the elderly represented the 65 or more segment, which is aligned with authors’ previous research and the findings of research-based user interface design authorities [46,47].

To recapitulate how demography-related aspects suited the overall approach, the conceptual model of the experiment to substantiate the FCD approach envisions that users who we focus on through the lens of FCD generations (adults and the elderly) participated in adaptive interface testing that respects equitable guidelines (WCAG 2.1 and others), which could provide a statistically significant difference between the original eDavki interface and an interface enhanced using the FCD-based approach for both of the generations included.

#### 3.2.1. Web Content Accessibility Guidelines 2.1

At the conception of the FCD interface variants, there was an imperative to follow established guidelines to accommodate a design strategy that would objectively contribute to users’ comfort. As there is a plethora of approaches when designing the visual interfaces for distinct generational groups, which are a focus in FCD, there were some reservations as to which would benefit the experiment. The scope was limited to well-established sources for more qualitative approaches, and the open standards of the W3C—the World Wide Web Consortium [48]—for a quantitative, programmable approach, where applicable. 

The Web Content Accessibility Guidelines (WCAG) 2.1 [7], published by W3C in 2018, is a successive document that expands upon WCAG 2.0., published in 2008, which addresses user interface design approaches that are nowadays common, in order to extend and clearly define criteria in overcoming accessibility issues in different types of internet content—primarily focusing on web pages. 

Accessibility guidelines address content on different terminals, from tablets, desktop to mobile and others, and are aimed primarily at a very wide range of disabilities, including accommodations for blindness and low vision and cognitive limitations, conditions that can affect all people through natural aging. “These guidelines also make web content more usable by older individuals with changing abilities due to aging and often improve usability for users in general” [7].

WCAG 2.1 follows four basic principles:Perceivable—information and user interface components must be presentable to users in ways they can perceive with their senses.Operable—user interface components and navigation must be operable by interaction the user can perform.Understandable—information and the operation of user interface must be within the limits of the user’s comprehension.Robust—content must be robust enough that it can be interpreted reliably by a wide variety of user agents, including assistive technologies available both at present and future technological progress.

To achieve a certain level of accessibility conformance as a subset of these principles, an array of 12 guidelines includes defined criteria that have to be met. Presently, the A-level, which is easier to maintain, even if met by 25 criteria, is considered below the good accessibility threshold; the AA-level consists of an additional 13 success criteria, while the AAA-level is progressively harder to achieve with an additional 23 criteria and is usually reserved for specialist solutions intended for people with disabilities. 

In this experiment, it was applicable to partially follow these guidelines, focusing exclusively on the visual presentation guidelines and success criteria due to the massive amount of web pages used within the testing portal interfaces—the visual, content information and layout structure could be improved, while the need to support time-based media (audio and video) alternatives to website elements was not foreseen. The specific criteria solving some of the age-related sensory and cognitive deterioration-related problems were intended to be to overcome with the enhanced visual experience of interfaces being supported by sources available on the W3C website [49]. Age-related functional limitations are accompanied by vision decline caused by changes in the physical condition of the eye. Often, this includes: the yellowing of the eye’s lens, and the loss of elasticity of the lens due to pupil shrinkage, resulting in a decreased capacity to focus, changed color perception, less light sensitivity, limited contrast sensitivity, and a reduction in the visual field [50].

Although some other criteria were met in the interfaces (e.g., multiple ways to target content), the success criteria that were pursued within the Perceivable principle, and could be detected in user interfaces via a programmable approach, are:1.4.3 Contrast (AA level) [51]. Text (including images of text) has a contrast ratio of at least 4.5:1. For text and images that are at least 18 pt, the contrast ratio is at least 3:1.1.4.6 Contrast enhanced (AAA level) [52]. The visual presentation of normal size text and text in images has a contrast ratio of at least 7:1; large text (more than 18 pt) or images have a contrast ratio of at least 4.5:1.1.4.12 Text Spacing (AA level) [53]. Criteria consist of several text presentation attributes: line height (line spacing) of at least 1.5 times the font size, spacing following paragraphs of at least 2 times the font size, letter spacing (tracking) of at least 0.12 times the font size, word spacing of at least 0.16 times the font size.1.4.8 Visual Presentation (Level AAA) [54]. The key website elements must be defined by several points, among them: width is no more than 80 characters or glyphs, non-justified text alignment, line spacing (leading) is at least a space-and-a-half within paragraphs, and paragraph spacing is at least 1.5 times larger than the line spacing.

#### 3.2.2. Testing Interface Definition

In a systematic overview of articles on the design of user interfaces for the elderly, difficulties for the elderly included [55]: physical issues, cognitive issues and computer experience, and the solutions typically came in the form of interface and control design, natural language, cognitive evaluation and input control adaptations. In terms of testing the user interface with the tool for remote testing, the interfaces were adapted to better support the solutions for some issues, but could not cover some components very effectively due to the nature of interfaces tested or due to the technical design of the testing tool. General usability and user-experience improvements are subject to interface and control design to provide a degree of text and object standards, intuitive control elements, with confirmations and errors and contextual help, where needed. The natural language that the user uses indicates the extent of complexity in certain processes they take part in, yet this entanglement is not necessary. It can be related to poor computer experience and understanding of certain processes. According to the systematic literature review digest [55], distinct aspects were identified to be addressed in user interfaces related to the elderly with physical (vision deterioration) and cognitive (attention, working and long-term memory) issues that needed enhancement.

To improve an important aspect of the FCD concept concerned with different levels of digital competencies, physical and cognitive issues among generations that can be included in the central point of interface development, the WCAG 2.1 success criteria were employed. This would provide a standardized approach toward adapted FCD-based interface conception, to avoid concept-specific solutions with new factors, which are difficult to avoid within the setting of the experiment.

At the experiment conception phase, the initial versions of both FCD interfaces, which were based on the AA- and AAA WCAG 2.1 success criteria, were shown to a small focus group that consisted of two adults and two elderly individuals to gather their first impressions and recognize critical problems the developers might have overlooked. Essential responses from the adult participants were “the AA version has good readability” and “the AAA version has too large fonts and not enough information on screen”, while the elderly argued that “the text in both versions should be even larger” and complimented the illustrations. The comments from the focus group and previous experience with adaptive user interfaces from research work on the FCD concept were the basis for the decision to implement two FCD-based interfaces, one for adults and one for the elderly.

The WCAG 2.1 criteria for key elements within the original eDavki interface were not met. The WCAG criteria met for the FCD interface for adults were strict, reaching the AA-level, while the FCD interface for the elderly met even stricter AAA-level success criteria. The core AA- and AAA-level success criteria enhancements were implemented in relation to text size, text spacing and color contrast in various website elements, such as content text, titles, navigation, etc., prescribed to comply with the four guidelines described in the former subsection. A specialized tool for WCAG 2.1 recommendations to inspect whether the chosen criterion has been respected on target interfaces was developed for the administration part of the platform, and considerably assisted in the interface development process. Apart from a programmable approach in developing WCAG 2.1-compliant elements of the interfaces, other, often subjective in nature, design decisions have been achieved by following good practices and rationale available in various sources, which are included in Table 1. For illustrative purposes, the choice of a violet color scheme was based on low arousal levels described in the color palette visual hierarchy post [56], while the most important elements, for example, submit buttons, were colored in contrasting, high arousal colors. Some of the practical solutions to these issues overlapped, e.g., text size and contrast colors of the theme, yet formed a distinguishable visual design. To test the usability of the overall FCD approach, the implementation of an adaptive user interface was proposed for these characteristics based on WCAG Guidelines (from the preceding section), or other recommendations indicated in Table 1 and visually represented in Figure 1. 

To sum up, all test participants would solve a task scenario in two interfaces: the original eDavki interface (labeled as Interface 1 in the Results section), and the adaptive, FCD-based variant (labeled as Interface 2 in the Results section). In the following variant, one of the two options was displayed: the adult or the elderly version, according to the generation classification of the user based on the demography questionnaire that the users completed at the beginning of the testing process. Table 1 represents differences between the different variants of the testing interface. 

An important remark: at the design phase of the FCD interfaces, scenario-specific target action funnels were actively avoided; although some enhancements offered some degree of simplification (e.g., around 20% fewer, non-relevant tax forms in the selection list), these adaptations could handle other scenarios within the eDavki tax portal.

### 3.3. Testing Platform

After conducting research on relevant unmoderated interface testing tools that would correspond with the experiment design requirements, a decision was made to develop a dedicated automated testing tool for the purposes of interface testing [60] due to the usage of an array of standardized measurement instruments in the local language: effectiveness, efficiency, a set of user evaluation questionnaires on system usability, user experience and task load. All considered measurement tools were not combined in any of the online tools as most commercially available solutions pursue more e-commerce-oriented usage.

It was sensible to invest time and knowledge in establishing an infrastructure for effective remote testing of user interfaces, which would solve a number of other problems, such as automated processing of survey data without manual input and accurate measurement of user experience with standard approaches [24]. The potential to reach a larger sample of test subjects was evident. While there were certain methodological reservations, for example, it would not be possible to detect other, verbal and nonverbal, signals such as in laboratory testing, these reservations were balanced by the significant advantages of remote, unmoderated usability [61], user experience and usage load measurements.

Due to the particular needs regarding the sufficient capacity of customization and technical limitations [62] of the user interface of the eDavki portal, the key features of such a platform were identified:Unmoderated remote testing. Participants can solve tasks independently, without a person conducting the testing. It is crucial that they cannot go outside the “testing area”, and at the same time they must not find themselves in a hopeless situation that would prevent them from completing the tasks, or this must at least be detected and appropriately marked in the data.Testing was based on task solving and standard measurements related to task performance and time. The participants received instructions, and then solved a series of tasks.Interactivity and adaptability of interfaces. The captured interface images should work as in the original. The interfaces should allow for a full user experience, and, at the same time, it had to be possible to adapt the presentation to test-specific display and interaction mechanisms. In the first iteration, the solution for desktops, with screen, mouse and keyboard interfaces, was developed, but in the future, testing with mobile devices would be a significant improvement.Adherence to good practices and standard user interface testing procedures. For example, the effect of learnability needed to be eliminated [38], so the tested interfaces and tasks appeared in a random order. For testing, standardized metrics from the field of usability and user experience research were used.Privacy and compliance with current rules for the use of personal data were mandatory.Re-usability of the tool for other researchers and projects was necessary.

These starting points served for the conception of the original framework of the environment for remote testing. There were no static constraints, and the tool could be adapted to any potential new requirements in the future. The user testing results for the frontend and backend were analyzed within a focus group consisting of user experience experts, programmers, and researchers focusing on designing digital services for the elderly, and were updated accordingly.

#### 3.3.1. Testing interface Website Capture

A key component in performing an experiment with existing user interfaces was snapshots or website mirrors, which provided the same user experience as the original, but ran completely independently on a separate server infrastructure to avoid any privacy and cybersecurity issues. This enabled testing copies of interfaces to be produced, and the integration of new visual approaches in a test environment, enabling direct programmable comparisons between different interfaces. In order to obtain snapshots of existing websites, it was necessary to carefully choose the right method of execution as a number of technical and other challenges were encountered. In this case, original proprietary scripts, which were spiders for the automated capture of web interfaces [63,64], could be used, but it was assessed that the use of established solutions would be more feasible [65]. Special attention was paid to the robustness and courtesy of the web spider. Regarding robustness: while browsing websites, a spider can be caught in a spider trap. This can lead to endless indexing of the same pages within a domain. Not all pitfalls are necessarily malicious, complications can also occur if the website has not been designed properly. As for the courtesy aspects: web servers contain various policies and recommendations that should be taken into account when visiting and capturing these pages with an automated script. Even if a website does not have a defined policy for web spiders, it is advisable to stick to good practices [66,67] that include: (i) A web crawler should present itself as a web crawler and not pretend to be an organic, human user. Websites record the number of visitors, and this allows them to control the bandwidth dedicated to web spiders. (ii) The web crawler must follow the rules of the robots.txt file, which determines which pages the crawler has access to. This allows operators to indicate which pages they do not want to have web crawlers access. In effect, this file does not prevent the spider from gaining access and can thus ignore it; nevertheless, care should be taken when accessing such websites as they may contain information that is prohibited from being stored (personal data and copyrighted works). (iii) A web spider should not consume too much bandwidth. This means that it does not transfer more than a limited set of files in a time period.

For the purpose of making snapshots of existing websites, a suitable existing tool [68] that would correspond to the specific needs of capturing the government tax portal with essential content after the login mechanism was sought. Special attention was made regarding the possibility of exporting the appropriate cookies to access private parts of web portals after logging in with a username and password. Of all the commercial and open-source tools tested, the Httrack open-source tool [69] proved to be the most suitable as it offered the widest range of possible settings while producing useful snapshots that worked well offline, offering identical user experience as the original website.

A formal permission from the operator of the eDavki portal to obtain a snapshot of the user interface for the needs of the research, within which we tested the applicability of the concept of family-oriented development, was granted.

There were more than 4500 files, and around 2200 of those were html content sites captured on the eDavki portal, which included generated copies of pages accessed by the web crawler via different routes. Thus, there was redundancy, yet this was not critical for the implementation of the experiment. Extensive work has been conducted to remove privacy-related information, links or form scripts that might be linked to the eDavki website and, thus, allow for testing-area exits and possible cybersecurity related incidents. This sanitized content served as a basis for the improved FCD-approach versions of the interface. The layout was optimized for desktops, with the aim of expanding to mobile in future research.

#### 3.3.2. User Testing Environment

After setting up the testing environment, which included the definition of user target actions, the input of instructions for each task, the removal of links or script calls that might reach outside the testing environment, and other parameters related to the interface appearance and behavior, the participants were invited with a single unified URL that was included in the invitation text and shared via social media, email and real-time chat communication. The invitation was disseminated in accordance with the objectives of the research and desired target sample.

Users who followed the link were shown the initial instructions to the objectives and purpose of the research, the person responsible and the data set that was being collected. After the user’s permission was obtained, the data collection started with the capturing of the start time, the Internet address (IP address), and the signature of the device and browser (through a browser/device fingerprinting technique) to prevent frequent attempts from one device. No explicitly personal information was saved (Figure 2a).

Next, the user submitted a simple demographic questionnaire (Figure 2b) that relates to the objectives of the survey, followed by the instructions for solving a series of tasks within the defined task scenario (Figure 3a). A description of each task was displayed in the interface. The task was considered successfully solved if the user performed the target action—e.g., a click on a certain object, which could be located anywhere in the interface structure. If an arbitrary time passed from the beginning of the task, the user was offered the option to skip the current task, and the system marked it accordingly—in this way, it was possible to increase the user’s ability to continue testing, thus providing a larger data set.

At the end of each set of tasks in each of the interfaces, standard questionnaires on user experience, system usability and workload were sequentially displayed (Figure 4). This was followed by a thank you message and updating of the database with the completion time of the testing. In addition, a 5-point scale for satisfaction with this remote testing tool and input for possible comments was displayed (Figure 3b).

#### 3.3.3. Backend Administration

Testing results were cumulatively processed and displayed in the backend administration interface intended for the rapid analysis of multiple frontend interfaces. Different sections addressed standardized measurements in user experience and usability research for the measured data (time on task, success rate, etc.), and subjective assessments (user experience, usability and task workload) of tested users within discrete interfaces through statistical methods for testing hypotheses based on distinctions in the usage behavior. To complement the user-test data, a recommendation tool for the analysis of appearances in the structure of distinct test interfaces was conceived in order to provide objective understanding of differences between disparate interfaces. It was not necessary to perform all the analyses in this interface; raw data could be exported and a statistical—or other types of—analysis could be delivered in statistical package programs, among others.

Please note that the display of results was generally adapted to the needs of the current experiment and interfaces based on the eDavki portal. Nevertheless, the processing and analytical methods and outputs could be freely customized to fit the objectives of specific testing requirements.

In addition to the testing variables’ setup and raw data table output, important sections of the administration interface consisted of the following:The dashboard included information on the experiment, describing the overall testing and did not fit into specific methodological categories in research analysis. This includes but is not limited to: the start and closing time of the experiment, overall cumulative of tasks finished and skipped, total time and clicks on tasks for each interface, cumulative questionnaires collected, and interface testing order. Further, the dashboard provided information on user-terminal characteristics: screen size, the browser and the operating system, mobile phone access attempts, and terminal fingerprint uniqueness that could identify multiple entries from a single device (even if accessed from different networks).The demography section was essential for conducting the experiment with FCD supported interfaces. This section provided information about users’ responses in demography surveys on their gender and age, if disclosed. It presented the classification for generation, adult or elderly, based on the developmental tasks of each generation the users ascribed themselves to.The measured responses provided the inputs from user task performance-related metrics to compare efficiency and effectiveness through time on task and success rate. The performance metrics were compared with a t-test to help us determine if the means had statistically significant differences. Although the number of clicks to accomplish each task was not considered as a measure comparable to standardized approaches to effectiveness or efficiency, it provided an additional perspective on performance of each task and scenario for distinct interfaces.Questionnaire responses (Figure 5a) comprised results on user self-assessment of usability (SUS—System Usability Scale questionnaire), user-experience (UEQ – User Experience Questionnaire), and task workload (NASA-Task Load Index questionnaire). The usability scale was presented as a unified measure of a subjective notion of the usability of the system, while the other two are composed of six discrete components (for UEQ: attractiveness, perspicuity, efficiency, dependability, stimulation and novelty), or factors (for NASA-TLX: mental demand, physical demand, temporal demand, own performance, effort and frustration). The results of each interface tested were collected and compared via a t-test for statistically significant differences in means. All questionnaires and components are thoroughly described in the Variables section below.Recommendations (Figure 5b) complemented the insights into user behavior within the testing environment with the interface characteristics analysis following the WCAG 2.1 documentation and other relevant guidelines that could be captured in an automated, programmable manner. There was a degree of novelty in non-trivial and profoundly crafted mechanisms to compare contrast, position, spacing, layout complexity, text size and colors of diverse elements on websites, and provide simple and specific recommendations if success criteria were not met (e.g., increase contrast, increase text size, etc.). Currently, there are some limitations in implementation due to specific requirements that focus on objective criteria in the visual representation of elements and semantically unstructured test-interface layouts. These mechanisms could be extended in the future with other non-visual criteria or automated user-interface web-mining functionalities [70] to retrieve “complexity signatures” on a large amount of pages that are part of a web portal.

## 4. Experiment

The following section provides information about the method, participants sample, and variables used to implement the experiment.

### 4.1. Method

So far, the distinct components of the testing platform were thoroughly described. The testing method demanded that all parts of the process required to achieve the desired outcomes were palpably described in order to explore specific research questions:Are there statistically significant differences in performance and user interface evaluation between the FCD-developed and the original eDavki interface?Are there statistically significant differences in performance and user interface evaluation between adult and elderly users?

On the basis of the research questions, it had to be determined whether there was a statistically significant difference in (1) which interface was “better” for both generations, and (2) if any generation was “better” in using both interfaces. The differences between generational groups as well as the differences in the use of interfaces were addressed; thus, a mixed factorial study design (2 × 2 factorial table) was conjured.

After experimenting with various methods for the statistical analysis of differences between means of data samples, the researchers utilized an analysis tool to compile an overview of topographic features of data samples. Although some samples showed potential symmetrical and mesokurtic distributions that might resemble normal distribution, and some samples even conformed to normal distribution, the Shapiro–Wilk formal test for normality, conducted on each data sample throughout performance and user evaluation dimensions within each interface and each generation, revealed that the majority of samples were statistically different from normal distribution (Appendix A).

It was assumed that the data were not normally distributed in the majority of the samples; hence, a nonparametric equivalent of the two-sample t-test, the Mann–Whitney U test [71], was used. The Mann–Whitney test is used when the following assumptions are met: the dependent variable should be measured on an ordinal scale or a continuous scale, the independent variable should be two independent, categorical groups, observations should be independent, and observations are not normally distributed. Presumably, the test is more robust to the presence of outliers than the t-test. The Mann–Whitney U test’s compliance to the null hypothesis significance testing framework is described in the Results section below, before the tables presenting the outcomes.

The primary goal was to resolve research questions, yet there were additional interesting insights accompanying the testing platform results discussed in the article. The results tables and graphs from the interactive testing platform are available in Appendix A.

### 4.2. Participants

Being cognizant of previous research [3,4] and associated conducted surveys, three general age-based groups were considered within the FCD concept: youth (aged 17 or less), adults (aged between 18 and 64), and the elderly (65 and above). Although there are other, more detailed generation classifications, this general classification was employed to avoid granularity of categories and probable overlapping of generation development goals, and to cater to sample structure challenges and statistical analysis. Youth was not included in the sample since the eDavki portal is intended for adult citizens to interact with eGovernment services. Due to the nature of the established digital solution, the FCD concept was only partially applicable, predominantly the component regarding different levels of digital competencies in generations. The structure of the sample was segmented into the adult and the elderly group to suit the appropriate theoretical background and the needs of the research.

Through a stratified random sampling method, the population was sampled from two strata based on their age characteristics. Based on the population census by the Statistical Office of the Republic of Slovenia in 2002, the elderly (aged 65 or more years) represented 22.24% of all adults. As displayed in Table 2, the elderly represented 24.56% of the total units in the sample, so distribution aligned with the general population of adult Slovenians was achieved.

The invitation was disseminated throughout social media platforms and email lists reaching the adult internet user population, who were presumably familiar with the tax portal. Special attention was paid to attracting older participants. After inviting the participants via researchers’ organization newsletters and social media channels, the response was considerable, yet few elderly individuals were reached. Subsequently, an invitation that would reach elderly participants was sent to 9 organizations for seniors on the national and regional levels and to individuals the research team was familiar with, which had an immediate effect on the achievement of a satisfactory number of testing participants in this age group.

There were 152 responses in the 14-day testing period in April 2021 (see Appendix A). The requirement was to specify age (as an independent variable), otherwise the test could not be continued. After missing cases were disregarded, 114 cases were analyzed. Among 114 participants, 71 (62.28 %) were male, 40 (35.08 %) were female, and 3 did not select gender. The participants’ age distribution is presented in Figure 6, and the generation groups are separated into the 18–64 years old adult generation (86 participants, 75.43%), and the 65 or more years old elderly generation (28 participants, 24.56%) in Table 2. The youngest participants were 23 years of age at the time of testing, and the eldest was 85 years old.

Additionally, to contribute to the FCD concept, a comparison between the age-based classifications and the developmental-goal-set-based question set was made and revealed an overall 80.53% fit where the age-based generation was equal to the developmental goals target (Appendix A). The fit was better for the adult (83.53%) than the elderly generation (71.43%)—presumably due to some elderly still engaging in life developmental goals typically attributed to the adult population. This experimental approach to generation classification could be interesting in conditions where it is not possible to ask users about age; rather, we could assume their generation based on particular developmental goal interests they disclosed.

Terminals that the participants used: 74.34% used Windows, 17.11% Mac OS, and 8.55% a Linux-based operating system (Appendix A). Among those 114 cases, system fingerprint uniqueness was complete. There were no multiple entries from a single device. The most common screen resolution was 1920x1080 with 41 cases, and the calculated average viewport size was 1809 × 1048 pixels. For details, please see Appendix A.

On average, users marked their familiarity with the Slovenian eDavki tax portal on a scale of 1 (none), 2 (somewhat familiar), 3 (using sporadically) and 4 (using regularly), with a score of 2.8214; thus, most of them are familiar with the portal and use it occasionally (see Appendix A for more details).

### 4.3. Variables

This section describes independent variables constituting stable conditions that are a basis for systemic control in the testing platform, and dependent variables that are dependent on other factors measured and are subject to change as a result of experimental manipulation of independent variables.

#### 4.3.1. Independent Variables

Generation (nominally):Adults.The elderly.

A demographic questionnaire (Figure 2b) was created according to the requirements of the experiment and is related to developmental goals on the basis of which we group the test participants into groups by generation, based on developmental goals as defined by sources in the literature [36,72], and used at the beginning of the testing process.

The researchers defined two groups of adult citizens, namely eDavki portal users: adults and seniors (predominantly younger retirees who knew how to use a computer). In addition to age, participants in the demographic questionnaire indicated which developmental tasks they encountered, e.g., establishing a household, parenting, caring for a partner, caring for other family members, developing a social network and leisure activities, common goals and interests, stressful situations and mid-life crisis, job satisfaction and success at work, community integration and belonging, adaptation to physical change—for adults; and maintaining health, adjusting to retirement, pursuing leisure activities, remembering past experiences, and socializing with peers and family—for the elderly. Based on the dominant types of developmental tasks for the individuals assessed, we ranked them as adults or elderly. The demographic questionnaire was completed by all the participants behind the computer terminal, and the classification into distinct generations was reflected directly in the user interface.

User Interface (nominally):Original.Dedicated (FCD).

Testing was conducted in two web-based interfaces. The original, current web interface of the eDavki portal was transferred from the eDavki.si portal and equipped with program code adapted for performing a series of tasks with a mechanism for capturing the performance and effectiveness of these tasks.

A dedicated interface was created using the family-centered design-based approach. The development of this interface focused on reducing the overall information congestion, on relevant solutions that were tailored to generation’s needs, aimed at explaining the purpose, process and goal of actions, and minimizing transitions between actions. Additional enhancements were more clear navigation, better readability, and an appropriate color scheme and other techniques for efficient interface design aligned with WCAG 2.1 criteria.

The family-centered design interface offered 2 distinct views based on generation association. The adult view complied with WCAG 2.1 level AA specific criteria, and the elderly view conformed with level AAA on distinct criteria selected due to its capacity to be applied within visual representation on a computer screen.

Counter-balance design was required in a sequence of testing interfaces based on two different approaches. In doing so, we counteracted the effect of learnability, which would potentially display better results for the last tested interface, after the participant has already come to know one similar interface. Therefore, the sequence of interfaces where the participant solves the tasks was appropriately mixed, as in similar experiments [24]. The generated results report (Appendix A) showed that the original interface first start order was in 74 cases, and 78 for the newly customized interface. There was no need for compensatory action in the task sequence itself in a distinct interface since the tasks were linearly related.

#### 4.3.2. Dependent Variables

Performance metrics provided insight into the usability of the systems or interfaces. These were captured automatically, as part of a proprietary, purposefully programmed tool to utilize the testing of different interfaces that emulated production conditions as realistically as possible:Success rate, as a metric related to effectiveness, was measured through programming logic. A task was marked as successfully completed if the user succeeded in completing the final action of a specific task, for example, clicking on the designated element within the tested user interface (e.g., the submit button). Otherwise, the task was marked as unsuccessful and handled accordingly when analyzing the data. Although the task success rate, which provided insight into distinct sections of the page and the task scenario, was measured for every task, in the results table there is a calculation of average task completion within task scenarios for each interface or generation.Time on task related to efficiency was measured for each of the three tasks that were part of the task scenario. In the given interfaces, the time measurement begins when the user clicks a button to start the testing, right after the instructions on performing a task scenario. For each task, the time on task is measured during the task completion action. In case of inability to complete the task, the time information is marked as 0 and handled separately when analyzing the data.

Standardized questionnaires provided researchers with additional insights into the self-reported experience of interfaces used by the participants. The test participants completed the questionnaires directly after testing, after accomplishing three tasks in each of two interfaces. The questionnaires were sequentially presented on the screen of the device. These standardized questionnaires, presented in the cross-reference and translated into Slovenian, were used in accordance with the literature and previous research efforts:The System Usability Scale (SUS) questionnaire with 10 question items was displayed on the first panel (Figure 4a) after a series of tasks at the completion of each interface. The results were stored after the “Continue” button was clicked before moving to the next questionnaire. The results are calculated and displayed as a SUS score (0 to 100) later in the administrator tool.The User Experience Questionnaire (UEQ) with 26 items was displayed (Figure 4b) in an identical manner, and after moving to the next questionnaire, the data are stored and calculated for the six components in the administration tool.The NASA Task Load Index (NASA-TLX) questionnaire (Figure 4c) was displayed on the final panel before the completion of testing or moving to another testing interface, depending on the randomized interface start order. The six-item questionnaire is concerned with various dimensions of stress, demand or complexity that participants have been subjected to during the system testing. The factors are then calculated in the administration tool and presented on 0 to 100 rank scales.

### 4.4. Results

The developed testing platform generated the following results in terms of detecting statistically significant differences related to research questions to accommodate the null hypothesis formal approach (whole set of hypotheses explicitly not composed in this article).

The selected Mann–Whitney U test for nonparametric samples prevalent in data samples collected in the experiment compares a randomly selected value from the first group to a randomly selected value from the second group. The null hypothesis assumes that there is a 50% probability that an observation from a value randomly selected from one collection exceeds an observation randomly selected from the other collection. If the probability of the randomly selected value from the first group not being equal to the randomly selected value from the second group is considered statistically significant, the null hypothesis can be rejected. Further, if the probability of the randomly selected value from the first group being equal to the randomly selected value from the second group is considered statistically significant, the null hypothesis cannot be rejected.

The results compiled in tables below present the Mann–Whitney U test null hypothesis rejection (value “1”), implying statistically significant differences among groups of data on each dimension, or null hypothesis non-rejection (value “0”), indicating the absence of statistically significant difference among groups of data on each dimension.

#### 4.4.1. Results by Interfaces

Comparing the original and FCD-based interfaces made it possible to test for statistically significant differences (Table 3) on performance metrics and user-assessed questionnaire results to determine which interface performed, and is preferred, by both generations.

In terms of success rate, there were no differences between the two interfaces. The success rate consisted of three disparate tasks with a maximum average of 1 (success) and a minimum of 0 (target action not completed), with a maximum total cumulative score of 3, as shown in Figure 7.

The time-on-task metrics showed a statistically significant difference as users took more time, on average, in the original interface to complete a single task than in the FCD-based interface (Table 3). The distribution of the average time per task in a scenario within an interface is presented in Figure 8. The difference is derived primarily from the task of filling in the form data sheet.

The user-assessed system usability scale scores displayed statistically significant differences. The second, FCD-based, interface acquired better scores (avg 64.8), which, in terms of comparable percentile SUS rankings, would be considered borderline “good” [25] (Figure 9).

The user experience questionnaire results produced statistically significant differences on the Attractiveness, Efficiency, Stimulation and Novelty components. The FCD-based interface was noticeably “better”, particularly with Attractiveness and its Hedonistic qualities subset, the Stimulation and Novelty components, while differences in terms of Pragmatic qualities (Perspicuity, Efficiency and Dependability) were somewhat lower. Note that results for each component higher than +1 and lower than −1 imply benchmarks for further consideration [29]. In this instance, this was not the case (Figure 10).

The NASA-Task Load Index by each interface presented no statistically significant differences in almost on all task load dimensions except for self-assessed performance and frustration. NASA-TLX implied slightly lower user’s self-assessed performance and effort in the FCD-based interface (Figure 11).

#### 4.4.2. Results by User Groups

Comparing the adult user group, as Generation 1 (18–64 years old), and the elderly user group, as Generation 2 (65+ years old), allowed us to test for statistically significant differences (Table 4) of performance metrics and user-assessed questionnaire results to determine which generation performed better within both interfaces. Primarily, this aspect examined whether there were general differences in average interface usage and evaluation between both generations.

A statistically significant difference in the task performance success rate between the generations was observed, where the elderly had a lower average success rate (Table 4). From Figure 12, it can be deduced that these differences were evident in tasks that involved finding the target tax form document (the results for each task are available in Appendix A).

A statistically significant difference was clearly present in the time-on-task metrics (Table 4) between averages of all tasks, as the elderly, on average, needed more time to complete all of the tasks in both interfaces. The average time on task metrics distribution is displayed for each of the tasks in Figure 13.

There was no statistically significant difference present in System Usability Scale user evaluation (Table 4). Both generations’ average SUS scores were comparable (illustrated in Figure 14).

Similarly, another user assessment in the form of UEQ results produced almost no statistically significant differences on all the components for both generations except for the stimulation component, where the elderly assessed both interfaces’ stimulation to be higher (Table 4 and Figure 15).

Except for Frustration, all NASA-Task Load Index components exhibited no statistically significant differences (as seen in Table 4 and Figure 16). The elderly, on average, reported less irritation with both interfaces.

#### 4.4.3. Isolated User Groups and Interface Data

Further investigation on testing results provided us with the opportunity to interpret additional details by “isolating” a data set to single independent variable’s conditions (e.g., for a single interface or single generation). These additional results were not widely presented in this article; however, they extend the understanding of the given research questions. For detailed tables and graphs, see Appendix A.

Data isolated for the adult user group (Generation 1) showed (Appendix A) that they had a partial preference for the FCD-based interface (Interface 2) on account of UEQ’s attractiveness and hedonic (stimulation and novelty) qualities, while no other performance or user evaluation metrics showed a statistically significant difference.

Isolated data for the elderly user group (Generation 2) similarly showed (Appendix A) a clear, above the threshold, preference for the FCD-based interface (Interface 2) on UEQ results and a better SUS score, NASA-TLX and time-on-task performance, without a statistically significant difference in success rate between the two interfaces.

When isolating data for a single interface, it was possible to observe that adults had better performance metrics (success rate and less time on tasks) than the elderly in the original (Appendix A) eDavki interface (Interface 1), while user evaluation (SUS and UEQ) did not show statistically significant differences. In this case, NASA-TLX produced mixed results for the set of dimensions.

In the FCD-based interface (Interface 2), adults had a better success rate, yet there are no statistically significant differences for time-on-task performance between generations, which would be considered an improvement in regard to original interface results. The average SUS score was evidently better for the elderly user group (Generation 2), while UEQ and NASA-TLX produced results partially in favor of the elderly on certain dimensions (Appendix A).

Generally, taking into account the isolated results for particular groups and interfaces where differences in means were statistically significant, it can be concluded that:According to SUS, UEQ and NASA-TLX, the elderly assessed the FCD-based interface as better and performed better on the time on task metrics compared to the original eDavki interface.With the original eDavki interface, the adults had better performance metrics.The elderly performed better in terms of time on task (with no difference compared to the adults) in the FCD-based interface.The FCD-based interface’s SUS was evaluated to be higher by the elderly.

## 5. Discussion

The following chapter discusses the implications of the established methodological approach on the results of the experiment. In addition, it indicates the limitations encountered. Last but not least, future work regarding the testing platform is presented.

### 5.1. Implications of Study and Results

According to the results presented in the section above, the outcomes of the experiment in terms of the testing of user performance and the evaluation of disparate interfaces representing the eDavki portal pointed out that:In global comparison, the FCD-based, WCAG 2.1-enhanced interface exhibited better time on task performance metrics and higher SUS scores for all testing participants, while user-evaluated dimensions within UEQ and NASA-TLX varied, and no impact on the success rate performance metrics was demonstrated.Both user groups, in general, did not give different responses in the self-reported evaluation, yet there were differences in both performance metrics, as the elderly were less successful and took more time to complete the tasks.Exploring the differences among singular interfaces and generations provided more indications that the FCD-based, WCAG 2.1-enhanced interface is better suited for the elderly, especially with respect to time on task performance metrics not being statistically significantly different from the adult generation group.Comparing the original eDavki and FCD-based, WCAG-enhanced interfaces, the time on task performance metrics have frequently shown better results for the FCD-based interfaces. It is important to observe that the lists and the forms within the adapted interface were not redesigned in a manner that would allow shortcuts or bypasses to task target actions. The enhancements implemented were exclusively related to a better perception and understandability of the system. To illustrate by means of an example that was not part of the Results section since the clicks per task metric was not included: time on a particular task where users had to find a specific tax form on a list was lower for the elderly group in the FCD-based interface, where the form list was grouped by important life event categories (bottom of Figure 1b), although it required users to click at least one time more to expand this section than in the original eDavki interface, where the list of tax forms had a flat structure (bottom of Figure 1a). In this example, the FCD-based interface conformed to the Perceivable (providing better readability), and Understandable (with segmented content structure for more context) principles of WCAG 2.1.

Regarding the FCD concept, it was demonstrated that improvements of certain aspects for different generations, even with relatively narrow modifications in visual presentation, were possible, and measurable differences were achieved in terms of following the evolving accessibility standards in the field. Even though the results indicate the usefulness of some aspects of the approach described above, the key experiment results indicate that:To expound on Research Question 1 on the differences in performance and user evaluation between the interfaces: The researchers cannot assume that the FCD-based interface categorically performed better and was preferred by all users on all dimensions. However, it was evident that statistically detectable differences were present in mixed performance and user-evaluation metrics in favor of the second, FCD-based interface, especially for important time on task metrics, the System Usability Scale and the majority of UEQ results. Further, these results partially corresponded to the approach where the service delivery, structure and content of both interfaces were in essence identical, while the changes were more evident on the graphic user interface in conforming to the WCAG 2.1 criteria, yet had an effect on SUS evaluation and principally led to better attractiveness and hedonic qualities within UEQ.To accommodate Research Question 2 on differences in the performance and evaluation between both generations: there were distinguishable differences in average performance metrics (success rate and time on task) between generations in favor of the adults, while self-assessed evaluations produced no statistically significant differences, with the exception of the UEQ’s stimulation component and the frustration dimension of the NASA-Task Load Index.

Important reservations regarding the study are discussed in the next sub-section.

### 5.2. Limitations of the Study

Firstly, there are challenges in terms of leveraging the unmoderated remote testing approach implemented for this experiment and the collecting of additional verbal or visual signals from the participants in a moderated laboratory testing environment. The researchers recognized the benefits of potentially larger participants’ reach, which was important during the Covid-19 crisis, and focused on ease of use for all participants, many of whom are elderly with low levels of digital skills. In the past, the authors used other digital tools for UX evaluation, such as eye tracking using a web camera in combination with mouse movement tracking for identifying users’ visual focus, while also allowing for detection of non-verbal cues of fatigue, etc. Monitoring participants through a web camera is applicable in smaller user samples in moderated settings. Remote monitoring is only possible with digitally literate users, who know how to enable the camera and deal with the settings. Since many of the test participants were lacking digital skills, the perceived value and applicability of eye and mouse movement tracking was questionable. That is why such UX evaluation approaches were not involved in the technical features of the testing platform.

Some limitations of the study regarding its representativeness for the general population or tax portal users are derived from the test participants’ sample size and structure. An assumption that existing eDavki users represent a distribution similar to the general population of adult Slovenians cannot be made, due to the lack of demographic data for the production version of the eDavki portal. In addition, there was an assumption that the established eDavki tax portal was less frequently used by seniors. With the remote testing approach, the researchers wanted to attract as many participants as possible through promotion on research organizations’ social media channels. To compensate for attracting the elderly user group, a special promotion was conducted through national and regional senior clubs and organizations. The response for each has been adequate, attracting 86 adults and 28 elderly citizens, thus achieving similar distribution of experiment participants as in the general population. Caution should be applied regarding whether this is a representative sample of actual eDavki users in terms of user behavior in this digital e-government solution.

Participation in the eDavki portal testing was appealing for the broader public as it is a well-known and widely used government public service portal. However, it should be noted that this experiment was not intended for in-depth changes in the delivery of core services of the eDavki web portal that would comprehensively address the necessary conceptual, technical and user aspects of modern e-government applications. Rather, it was focused on testing general recommendations and standards used in the implemented FCD concept, and validation of the platform for unmoderated remote testing to produce general guidelines for improvements of online services. To effectively improve the existing eDavki portal, a longer, more in-depth, more complex and interdisciplinary approach would be required, including tax, public services, and law expertise, among others.

While the existing public digital tax portal was chosen due to its relevance for citizens’ everyday life and popularity among digital services, which helped attract a wider number of participants, it is not applicable to test every FCD concept principle described in the Background section. Particularly, the part addressing common family activities is challenging since the tax portal is tied to adult citizens’ identity and limited to scenarios and interaction with government institutions; thus, it leaves out the younger generations, and in essence provides more observation into elderly-specific UX challenges and improvements than family-related dynamics.

Throughout the development of the FCD concept, it was the research team’s ambition to achieve a meaningful impact in digital service delivery, not just toward the individual user, but to include groups of users that have distinct requirements. In previous research efforts, the focus was on group processes and service delivery structure, which is implemented in the MyFamily application prototype [3]. The design and development of the MyFamily prototype and the experimentation with users followed the FCD concept. For the latest research, presented in this paper, it was necessary to test the transferability of the concept to non-proprietary and existing digital services, such as eDavki. Initially, this posed a small challenge as, if the customization of the second interface was conducted solely according to arbitrary user interface design decisions and user engagement strategies, this research would have had a very limited impact. Testing the differences in such interfaces could provide more experimental freedom and could immediately display better performance and user evaluation results, though limited to particular interfaces and dangerously exposed to an array of factors that are too complex to analyze properly. The quantitative methods to explore research objectives are vitally important; the authors conclude that the analytical, statistical framework would be better suited if researchers exercised a narrower, yet pertinent analysis of factors, not relying on previous research work and examples in similar UCD methodology-based articles. Taking into account of the complexity and transferability limitation issues, certain decisions related to the interface and testing platform were enforced: restricting the testing to desktop devices only, implementing standardized user interface testing methods used in previous stages of concept development, enhancing visual presentation only on accounts congruent with evolving accessibility web standards (i.e., WCAG 2.1), limiting the complexity of task scenarios and retaining the large structure of the original interface, captured with a specialized webcrawler mechanism.

### 5.3. Testing Platform and Future Development

An important component of the FCD concept evaluation was the design, coding and testing of the platform aimed at unmoderated remote testing. While standard performance and user evaluation instruments were already used in previous experiments with the proprietary MyFamily web application, the new platform was essential for testing other interfaces. The new platform for unmoderated remote testing resulted in an enhancement of the testing process due to the larger participants’ sample size and more diverse structure, shortened time for moderation and automatic compiling of results. In addition, it enabled testing format that can avoid the health-related risks of current and future pandemics. Due to the platform, it was possible to move from a live laboratory environment, where only the most willing participants attend the tests, and thus typically providing higher scores on subjective evaluation instruments when interacting with the testing moderator. All these enhancements were not self-evident and required some focus group engagement to implement a safe, anonymous, comprehensible, wholesome, non-irritating online interface testing environment, and to enable participants to skip tasks they are not able to resolve without stopping the testing of other parts or interfaces, thus providing as much relevant data as possible. There were additional insights into the commentary and evaluation of the testing process from an optional form displayed at the end of the testing process. It is acknowledged by the research team that the scores and answers given there were only from the most interested participants, though they provided valuable confirmation regarding the testing process and testing environment. Summarizing the comments, there were some that expressed a wish to implement changes in the existing eDavki portal (which was not the goal of this research), some warned about the learnability effects (which were resolved with counter measures), and some commented on inconsistency, and lack of real change in the service delivery of interfaces (which is correct as varying amounts of visual changes were implemented). None commented on the quality of the instructions, testing environment or visual problems. Overall 52 users that marked the testing process satisfaction score from 1 (not satisfied) to 5 (very satisfied) gave an average rating of 4.46 (Appendix A), which is encouraging for the future development of the platform.

Based on the findings of the experiment, the authors conclude that the interactive unmoderated remote testing platform served the purpose of experimenting with the FCD concept with WCAG 2.1 and other enhancements. The testing environment, together with the existing website capture mechanisms and administration components for compiling results and recommendation tools, is useful and transferable to other future performance and user evaluation experiments, as it was based on standardized testing methods and automated web accessibility standards verification. Future evolution of the testing results’ statistical compilation tool will provide formal tests for normality of sample distribution and automated identification of methodologically sound tests to provide immediate statistically significant inferences from the data collected.

The inherent transferability potential of the development approach makes it possible to inaugurate the testing platform on basic LAMP (Linux, Apache, MySQL and PHP) web hosting infrastructure. The widely popular PHP, HTML5 and JavaScript web programming stack ensures a conventional object-oriented programming approach with proper customization capabilities. The key investment of using the proposed platform for unmoderated remote testing includes time for the preparation of testing interface(s) and testing tool construction according to an original experimental design, with a prerequisite knowledge of UCD and statistical methodology employed. In order to reuse the testing platform for other similar user performance and user interface evaluations, third parties, e.g., researchers, are encouraged to contact the authors about the instructions regarding installation, modification, the sharing of good methodological practices, and consultation on their own experimental set-ups.

## 6. Conclusions

This article begins with a revisiting of the background and research techniques of UCD complemented with the FCD concept developed in the previous iteration [3,4], where this concept evolved from determining the role of the family in the context of Smart City digital services and associated generational characteristics using a distinct interdisciplinary approach, followed by the practical design and development of a corresponding testing framework, with interactive, remote, unmoderated, automated query platform features.

Effectively, a webcrawler-captured mirror image of the existing (original) eDavki portal, including the private access section with a vast amount of subpages and a customized interface that was similar in document structure and content, but with enhanced, relatively narrow, yet visible differences in presentation based on the generational needs of the FCD-concept approach and the W3C Consortium’s Web Content Accessibility Guidelines (WCAG) 2.1, that could be verified by the testing platform’s recommendation tool, was tested. User performance (task success rate) and efficiency (time on task) were measured on a predefined set of tasks and user evaluation questionnaires—a user experience questionnaire, system usability scale, and NASA-Task load index—while aggregating other useful data.

To synthesize research questions, results compiled from the backend analytics interface confirmed that there are observable statistically significant differences among several global interface usage performance factors and user evaluation dimensions. Further examination of data separated for single generations and single interfaces indicated that the elderly benefit performance-wise from the FCD-based interface, specifically in terms of lower time-on-task metrics. Although testing the eDavki interface produced partial results in some instances, the FCD approach as a part of broader UCD methodology could contribute to improved user experience and usage performance for all generations through a further iterative development process.

A meaningful aspect to consider in the future development of remote unmoderated user testing platforms is to rectify the code in order to improve the testing platform data presentation, analytics and recommendation tools, and to compose a single package that can easily be transferred to third parties and used for different conditions in user interface testing using standard approaches.

The FCD concept approach will further strengthen the established UCD process in our endeavors to promote and support digitalization, notably in the field of common public services, identifying and addressing deficiencies in the societal topology related to digital competencies and technology access, and thus improving the quality of life of citizens, families and other closely connected groups in the modern digital era.

## Figures and Tables

**Figure 1 sensors-21-05161-f001:**
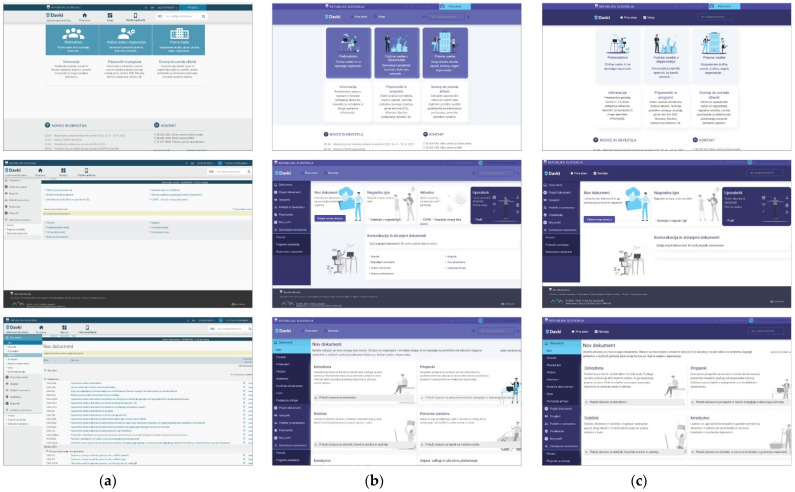
Visual presentation of (**a**) original, (**b**) FCD adult and (**c**) FCD elderly implemented interfaces.

**Figure 2 sensors-21-05161-f002:**
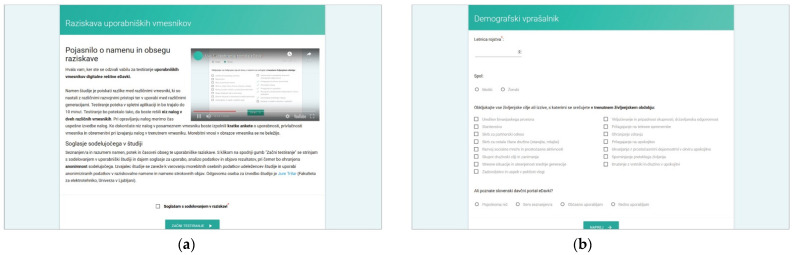
Testing platform interface displaying (**a**) introductory text with user agreement dialog, and (**b**) demography, generation developmental task and familiarity with the solution questionnaire.

**Figure 3 sensors-21-05161-f003:**
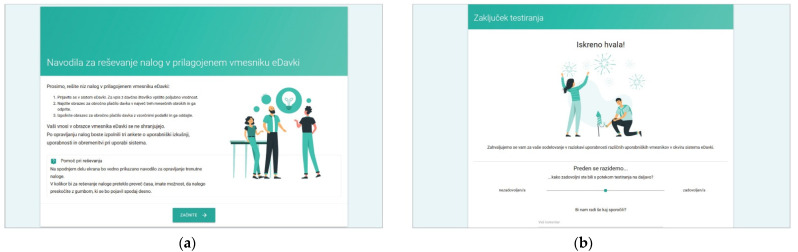
Testing platform interface displaying (**a**) the users’ task instructions and (**b**) the final “thank you” screen.

**Figure 4 sensors-21-05161-f004:**
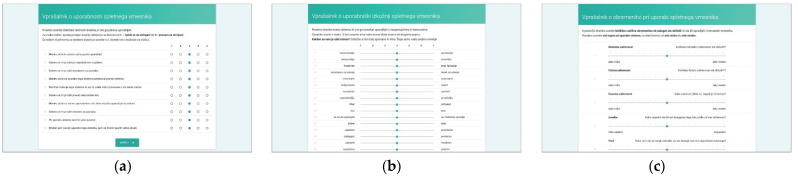
Testing platform interface displaying (**a**) SUS, (**b**) UEQ, and (**c**) NASA-TLX questionnaires.

**Figure 5 sensors-21-05161-f005:**
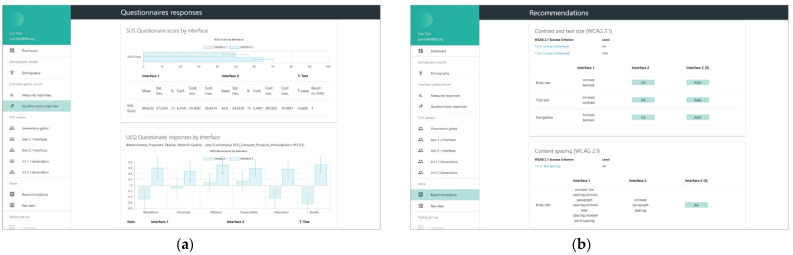
Backend administration displaying (**a**) user evaluation results, and (**b**) WCAG 2.1 recommendation tool.

**Figure 6 sensors-21-05161-f006:**
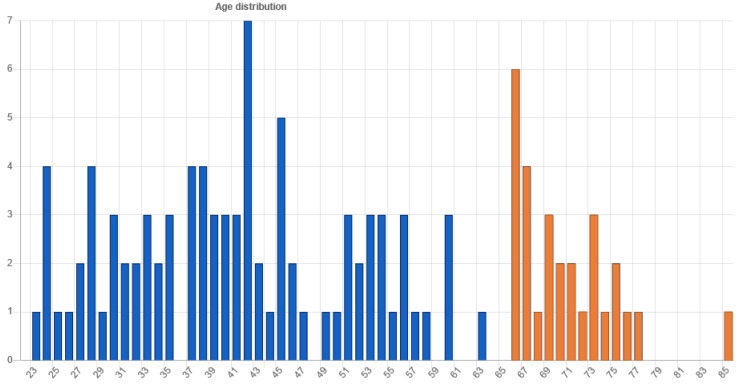
Participants’ age distribution.

**Figure 7 sensors-21-05161-f007:**
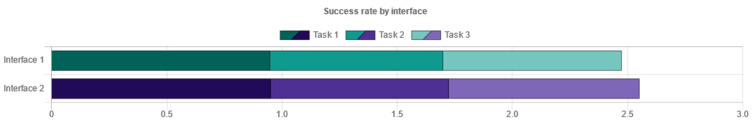
Success rate by interface.

**Figure 8 sensors-21-05161-f008:**
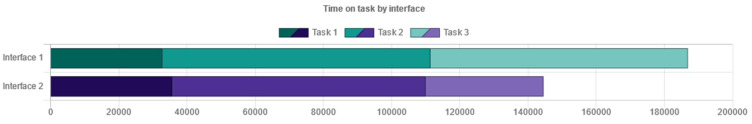
Time on task by interface in milliseconds.

**Figure 9 sensors-21-05161-f009:**
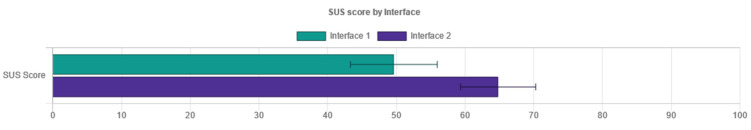
SUS score by interface.

**Figure 10 sensors-21-05161-f010:**
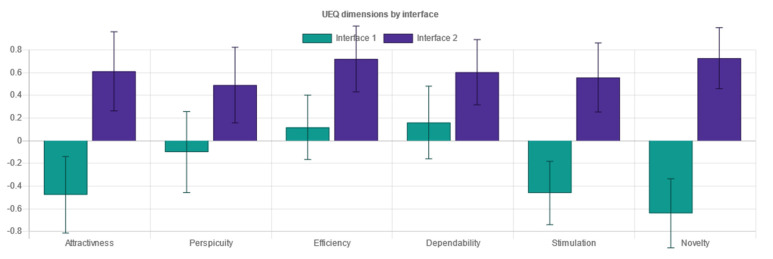
UEQ dimensions by interface.

**Figure 11 sensors-21-05161-f011:**
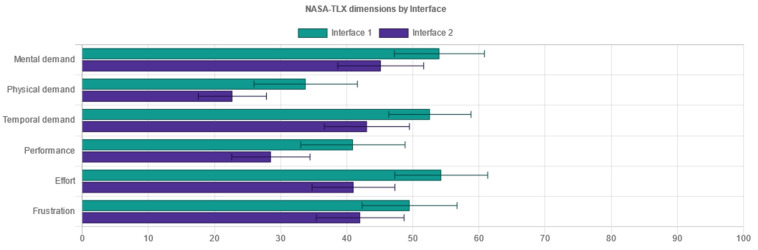
NASA-TLX dimensions by interface.

**Figure 12 sensors-21-05161-f012:**
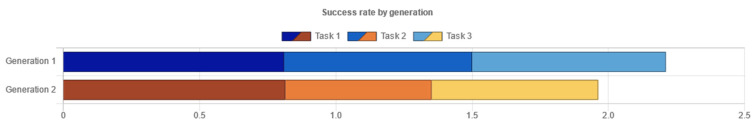
Success rate by generation.

**Figure 13 sensors-21-05161-f013:**
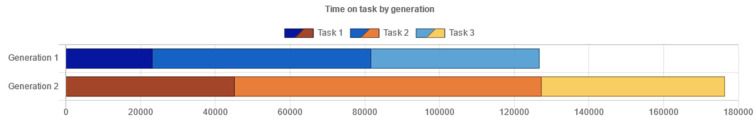
Time on task by generation in milliseconds.

**Figure 14 sensors-21-05161-f014:**
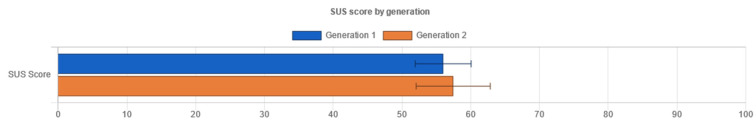
SUS score by generation.

**Figure 15 sensors-21-05161-f015:**
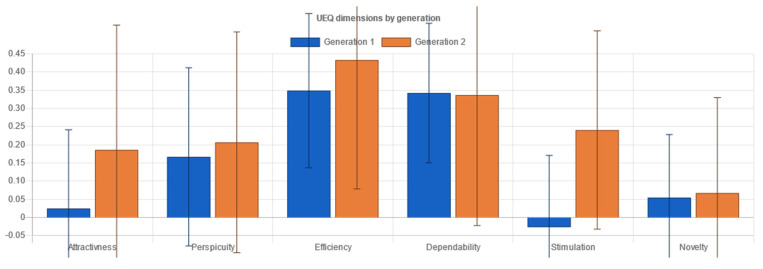
UEQ dimensions by generation.

**Figure 16 sensors-21-05161-f016:**
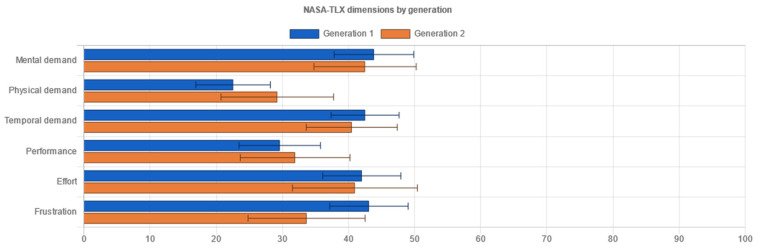
NASA-TLX dimensions by generation.

**Table 1 sensors-21-05161-t001:** Differences between the original, FCD adult and FCD elderly testing interfaces.

Factor	Original eDavki Interface (Interface 1)	FCD Generation-Based Adaptive Interface (Interface 2)	
		FCD Adult	FCD Elderly
**Color scheme**	teal, dark teal, gray, white	low arousal colors [56]: violet palette, light blue, white	low arousal colors, partial dark mode [56,57]: dark violet, violet, light blue, white
**Key elements color**	dark teal	red, contrasting to default color palette [58]	red, contrasting to default color palette [58]
**Default font**	Lato	Roboto [59]	Roboto [59]
**Body text**	WCAG 2.1 text size and contrast: does not meet WCAG 2.1 spacing and presentation: does not meet	WCAG 2.1 text size and contrast: AA [51] WCAG 2.1 spacing and presentation: AA [53]	WCAG 2.1 text size and contrast: AAA [52] WCAG 2.1 spacing and presentation: AAA [54]
**Title text**	WCAG 2.1 text size and contrast: does not meet WCAG spacing and presentation: does not meet	WCAG 2.1 text size and contrast: AA [51] WCAG 2.1 spacing and presentation: AA [53]	WCAG 2.1 text size and contrast: AAA [52] WCAG 2.1 spacing and presentation: AAA [54]
**Navigation**	WCAG 2.1 text size and contrast: does not meet WCAG 2.1 spacing and presentation: does not meet	WCAG 2.1 text size and contrast: AA [51] WCAG 2.1 spacing and presentation: AA [53]	WCAG 2.1 text size and contrast: AAA [52] WCAG 2.1 spacing and presentation: AAA [54]
**Buttons**	original size	larger (wide target area)	enlarged (wider target area)
**Content sections**	squares	neomorphic cards	neomorphic cards
**Image content**	pictograms	illustrations	high contrast illustrations
**Forms list**	robust list of all forms	segmented lists based on tax type/life events, all forms available	segmented lists based on tax type/life events, excluding 20% forms not needed for seniors
**Content and language**	formal, certain parts lack explanation, certain parts with “wall of text”	added explanations for context	added shorter explanations for context

**Table 2 sensors-21-05161-t002:** Age and generation.

	N	%	Min	Max
All (without missing)	114	100%	23	85
Generation 1 (18–64)	86	75.43%	23	62
Generation 2 (65+)	28	24.56%	65	85

**Table 3 sensors-21-05161-t003:** Performance metrics and user evaluation for testing interfaces.

		Interface 1				Interface 2				Mann–Whitney U Test	
	Dimensions	Mean	Std. Dev. ^1^	N ^2^	Conf. ^3^	Mean	Std. Dev. ^1^	N ^2^	Conf. ^3^	*p*-Value	H_0_ Rejection (α = 0.05)
**Performance metrics**											
**Success rate**		0.8250	0.3808	240	0.0482	0.8506	0.3572	261	0.0433	0.4380	0
**Time on tasks** (**ms**)		62,304	112,585	240	14,244	48,186	61,526	261	7464.4660	0.02112	1
**User evaluation**											
**SUS**		49.6233	27.5242	73	6.3141	64.8000	24.1678	75	5.4697	0.001044	1
**UEQ**	Attractiveness	−0.4762	1.4386	70	0.3370	0.6111	1.5396	75	0.3484	0.00004424	1
	Perspicuity	−0.1000	1.5230	70	0.3568	0.4900	1.4690	75	0.3325	0.05805	0
	Efficiency	0.1179	1.2082	70	0.2830	0.7200	1.2807	75	0.2898	0.002178	1
	Dependability	0.1607	1.3647	70	0.3197	0.6033	1.2688	75	0.2872	0.07301	0
	Stimulation	−0.4607	1.1900	70	0.2788	0.5567	1.3423	75	0.3038	0.00004006	1
	Novelty	−0.6393	1.2991	70	0.3043	0.7267	1.1878	75	0.2688	8.112 × 10^−9^	1
**NASA-TLX**	Mental	54.0000	29.0751	70	6.8113	45.1351	28.5251	74	6.4993	0.05798	0
	Physical	33.7857	33.3670	70	7.8167	22.7027	22.6224	74	5.1544	0.1228	0
	Temporal	52.5714	26.5633	70	6.2229	43.0405	28.2457	74	6.4357	0.05035	0
	Performance	40.9286	33.7004	70	7.8948	28.5135	25.9970	74	5.9233	0.05756	1
	Effort	54.2857	30.0035	70	7.0288	41.0135	27.4982	74	6.2653	0.00692	1
	Frustration	49.5000	30.6234	70	7.1740	42.0270	29.1068	74	6.6318	0.1599	0

^1^ Standard deviation. ^2^ Number of cases. ^3^ Confidence.

**Table 4 sensors-21-05161-t004:** Performance metrics and user evaluation of interfaces between generations.

		Generation 1 (18–64 Years)				Generation 2 (65+ Years)				Mann–Whitney U Test	
	Dimensions	Mean	Std. Dev. ^1^	N ^2^	Conf. ^3^	Mean	Std. Dev. ^1^	N ^2^	Conf. ^3^	*p*-Value	H_0_ Rejection (α = 0.05)
**Performance metrics**											
**Success rate**		0.7371	0.3521	213	0.0473	0.6543	0.3924	81	0.0855	0.01021	1
**Time on tasks** (**ms**)		42,260	61,400	213	8245	58,806	65,042	81	14,164	0.003251	1
**User evaluation**											
**SUS**		55.9914	15.7832	58	4.0620	57.419	14.0200	26	5.3891	0.7861	0
**UEQ**	Attractiveness	0.0249	0.8330	57	0.2163	0.1859	0.8920	26	0.3429	0.5446	0
	Perspicuity	0.1667	0.9429	57	0.2448	0.2067	0.7889	26	0.3033	0.9331	0
	Efficiency	0.3487	0.8162	57	0.2119	0.4327	0.9214	26	0.3542	0.5707	0
	Dependability	0.3421	0.7382	57	0.1916	0.3365	0.9333	26	0.3587	0.7929	0
	Stimulation	−0.0263	0.7584	57	0.1969	0.2404	0.7088	26	0.2725	0.02848	1
	Novelty	0.0548	0.6673	57	0.1732	0.0673	0.6821	26	0.2622	0.5700	0
**NASA-TLX**	Mental	43.8393	22.9771	56	6.0181	42.5000	20.1370	26	7.7404	0.9801	0
	Physical	22.5446	21.5031	56	5.6320	29.2308	22.1559	26	8.5165	0.1205	0
	Temporal	42.5000	19.6561	56	5.1483	40.4808	17.9028	26	6.8816	0.7602	0
	Performance	29.5982	23.4946	56	6.1536	31.9231	21.5790	26	8.2947	0.6852	0
	Effort	42.0089	22.5349	56	5.923	40.9615	24.5772	26	9.4472	0.8692	0
	Frustration	43.0804	22.6284	56	5.9267	33.6538	23.0459	26	8.8586	0.03818	1

^1^ Standard deviation. ^2^ Number of cases. ^3^ Confidence.

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
