# Peer review of "Family-Centered Design: Interactive Performance Testing and User Interface Evaluation of the Slovenian eDavki Public Tax Portal"

_sensors, 2021, doi:10.3390/s21155161_

Round 1

Reviewer 1 Report

ear authors

The topic of the paper is interesting however there are some points that need to be addressed before publication.

  1. The paper is written to first person. Please re-write it in the third person. You made many self-citations explaining your previous works. It would be more preferably to refer to this works as you refer to work of others. For example refer to [3] as Trilar et al. [3] e.g Trilar et al. [3] focused predominantly on how to develop a different approach and prototype new.…

  2. At the end of the introduction please add the structure of the paper

  3. Regarding literature review. you use user experience (UEQ questionnaire), usability (SUS questionnaire) and task work-load assessment (NASA-TLX questionnaire), but you do not refer to them at the background. Section 2.2 User-centered design methodology & standards is very general. You refer for example to Nielsen regarding usability but you use SUS questionnaire for usability testing. So you have to be more specific and explain how SUS questionnaire was used in previous studies. Do the same for all of the instruments that you use.

  4. My main objection is about family-friendly approach. I can see adult and elderly testing interfaces. For example in table 4 you make comparisons between adult user group, as Generation 1 (18-64 years old), and the elderly user group, as Generation 2 (65+ years old). So, at the creation of your sample how did you incorporated the family concept?

  5. In table 1 You describe thee interfaces. The original one and the adult and elderly interfaces but later on in table 3 Performance metrics and user evaluation for testing interfaces you refer two interfaces. Why? I suppose that interface 2 refers to FCD-based interfaces. How have you evaluated the performance metrics?

  6. Generaly it was very difficult for me to follow your paper, especially the results section. Please make sub-sections to present your results.

Author Response

Dear Reviewer,

thank you for reviewing the manuscript. We greatly appreciate your comments and recommendations that help us improve the article. We have revisited the manuscript accordingly and made structural and content related changes. As there is a considerable amount of updates in the manuscript, while using “track changes” document mode we recommend to use the “Simple Markup” option for optimal reading experience.

The following is our point-by-point responses. We hope that you find our responses satisfactory.

---

The topic of the paper is interesting however there are some points that need to be addressed before publication.

Point 1: The paper is written to first person. Please re-write it in the third person. You made many self-citations explaining your previous works. It would be more preferably to refer to this works as you refer to work of others. For example refer to [3] as Trilar et al. [3] e.g Trilar et al. [3] focused predominantly on how to develop a different approach and prototype new.…

Response 1: The article was rewritten to expunge first-person pronouns scattered throughout the text. References to own previous articles (Trilar et al.) were added to emphasize the related work on central concept of this article.

Point 2: At the end of the introduction please add the structure of the paper

Response 2: The structure of the paper has been added at the end of introduction section (Line: 83).

Point 3: Regarding literature review. you use user experience (UEQ questionnaire), usability (SUS questionnaire) and task work-load assessment (NASA-TLX questionnaire), but you do not refer to them at the background. Section 2.2 User-centered design methodology & standards is very general. You refer for example to Nielsen regarding usability but youu use SUS questionnaire for usability testing. So you have to be more specific and explain how SUS questionnaire was used in previous studies. Do the same for all of the instruments that you use.

Response 3: The standardized UCD related instruments and questionnaires descriptions were moved to 2.2 User-centered design methodology & standards section from 4.3 Variables subsection and additionally commented (line 215).

Point 4: My main objection is about family-friendly approach. I can see adult and elderly testing interfaces. For example, in table 4 y0ou make comparisons between adult user group, as Generation 1 (18-64 years old), and the elderly user group, as Generation 2 (65+ years old). So, at the creation of your sample how did you incorporated the family concept?

Response 4: The simple generations classification was used due to the researchers’ previous work, also to avoid granularity of data thus better suitability for statistical tests conducted. In respect to generations participating (no youth) and established tax portal scenarios the FCD approach was only partially included in part concerned with different levels of digital competences among generations. These constraints were additionally addressed in Section 4.2 Participants (line 835) in regard to sampling and concerns were also expressed in the 5. Discussion section in relation to Limitations of the study.

Point 5: In table 1 You describe thee interfaces. The original one and the adult and elderly interfaces but later on in table 3 Performance metrics and user evaluation for testing interfaces you refer two interfaces. Why? I suppose that interface 2 refers to FCD-based interfaces. How have you evaluated the performance metrics?

Response 5: The authors can acknowledge that there is more clarity in this regards needed, so some additional explanation was added. The table presents all three variants, the second column merging the two adaptive interfaces under FCD based approach to interface design. (line 592). The differences between two tested interfaces was expounded in 3.2.2. Testing interface definition section (line 534). The original eDavki interface (No WCAG 2.1 conformity) was labeled as Interface 1 while FCD based adaptive interface, that provides one of two options, adult (WCAG 2.1 AA success criteria) and elderly (WCAG 2.1 AAA success criteria) that was labeled as Interface 2 in the results section. Due to design of the experiment, every participants performance metrics were measured in 1-original interface, 2-FCD (adult or elderly) interface – note that differences between both FCD-based interfaces were merely in slight color palette changes (better contrast) and text size and spacing. Additional descriptions of performance metrics and user evaluation instruments related to testing platform process were expounded in 4.3 Variables section in Dependent variables section (line 947). Other general performance metrics (effectiveness, efficiency) descriptions were added in relation to Point 3 under 2.2 User-centered design methodology & standards section.

Point 6: Generaly it was very difficult for me to follow your paper, especially the results section. Please make sub-sections to present your results.

Response 6: Authors are aware of longer and difficult manuscript structure as there were many aspects to consider and to cover as many potential issues as possible. Authors hope various amendments and overall structure improvements will offer more clarity. The 4. Results section remains structured as is, yet has been improved with color coded cues in barcharts for more clarity regarding distinct categories (e.g. light blue for adult generation, orange for elderly, cyan for original interface, violet for FCD interface). The 5. Discussion section has been revisited and segmented into subsections, the first being related to results where the authors try to offer more general insights into results implications.

Reviewer 2 Report

The aim of this paper is to determine if there are significant differences among the original interface design of the eDavki tax portal and the alternative designs for adult and elderly users the authors proposed based on the family-centered design (FCD) authors are introducing. For that purpose, the authors carried out an empirical study in which they employed application designed for remote testing of different dimensions of usability and accessibility. The application itself contains widely used measuring instruments such as UEQ, SUS, and NASA-TLX, as well as instruments for evaluating accessibility according to WCAG. The paper is well organized and written but I have several concerns related to the employed research methodology. The authors should clarify why elderly people constitute the representative sample of tax portal users since they were involved in the study as one of the essential group the study was focused on. When analyzing data, the authors used t-test but is not clear if the authors have tested the assumptions for using the t-test in the first place. Moreover, according to authors' description of the study, some assumptions have not been reached (measurement scale, random sample, and large sample size) which indicates that reported findings might be biased if the authors have used the parametric test when they have to employ its non-parametric counterpart. Given that the authors are dealing with the usability evaluation, more attention should be paid to the aesthetics of presented study results since combination of blue, green, and grey is not appealing to the eye. Section on study implications and limitations is missing as well. Considering the aforementioned, the authors are encouraged to rework their paper.

Author Response

Dear Reviewer,

thank you for reviewing the manuscript. We greatly appreciate your comments and recommendations that help us improve the article. We have revisited the manuscript accordingly and made structural and content related changes. As there is a considerable amount of updates in the manuscript, while using “track changes” document mode we recommend to use the “Simple Markup” option for optimal reading experience.

The following is our point-by-point responses. We hope that you find our responses satisfactory.

---

The aim of this paper is to determine if there are significant differences among the original interface design of the eDavki tax portal and the alternative designs for adult and elderly users the authors proposed based on the family-centered design (FCD) authors are introducing. For that purpose, the authors carried out an empirical study in which they employed application designed for remote testing of different dimensions of usability and accessibility. The application itself contains widely used measuring instruments such as UEQ, SUS, and NASA-TLX, as well as instruments for evaluating accessibility according to WCAG. The paper is well organized and written but I have several concerns related to the employed research methodology.

Point 1: The authors should clarify why elderly people constitute the representative sample of tax portal users since they were involved in the study as one of the essential group the study was focused on.

Response 1: In respect to generations participating (no youth) and established tax portal scenarios the FCD approach was only partially included in part concerned with different levels of digital competences among generations. To support the general digitalization efforts the researchers perceived that the need to improve access and usability of digital services for all generations (competence levels) is very important. In reality the elderly represent a very small portion of eDavki tax portal, yet the recommendation regarding certain aspects of the interface would gradually improve the number of elderly users. To test the differences among generational groups the researchers had to invite enough senior participants for at least some statistical relevance. Some of the above comments were included in 1. Introduction (line: 55), section 4.2 Participants (line 835), and some concerns were also expressed in the 5. Discussion section in relation to Limitations of the study (1166). Please, also see Point 3 about the sample representativeness. 

Point 2: When analyzing data, the authors used t-test but is not clear if the authors have tested the assumptions for using the t-test in the first place.

Response 2:  The authors used t-test assuming the bivariate independent variable (A, B groups) with ratio or interval scale on dependent variables, and measuring means on these factors. Following the guidelines on robustness on non-normality of random data/sample the 2-sample t-test condition that each group should have at least 15-20 units and based on the related work that is concerned with similar performance and user-evaluation comparison of means the authors decided that t-test would be applied. Of course the authors acknowledge the potential issues related to non-trivial statistical inference, so the 4.1 Method section was updated (line 822) and concerns expressed in 5.2. Limitations of the study section (line 1219).

Point 3:  Moreover, according to authors' description of the study, some assumptions have not been reached (measurement scale, random sample, and large sample size) which indicates that reported findings might be biased if the authors have used the parametric test when they have to employ its non-parametric counterpart.

Response 3: Firstly, thank you for pointing out the aspect on random sample importance - a mistake regarding the sampling method was updated (not expert selection rather it was stratified random sampling method as the researchers had no role in choosing participants), and assertion regarding the representativeness of the sample in relation to the general Slovenian population has been added in 4.2 Participants section (line 848). Additional thoughts on sampling techniques were added in the 5.2 Limitations of the study section (Line 1166). 

Please see authors response in connection to previous Point 2. Caution that there are additional considerations on the usage of independent and multiple dependent variables in comparing means in such experiment setting and testing platform analysis tool has been added (Line: 1219).

Point 4: Given that the authors are dealing with the usability evaluation, more attention should be paid to the aesthetics of presented study results since combination of blue, green, and grey is not appealing to the eye.

Response 4: The authors agree on the importance of visual presentation. Variation with more hue and saturation added. The colors now represent unified visual cues for testing interfaces- (cyan, violet) and generations- (light blue, orange) related data for more clarity. Figures updated throughout the manuscript and the supplement.

Point 5: Section on study implications and limitations is missing as well.   

Response 5: The 5. Discussion section (line 1107) has been segmented into relevant subsections related to present better structure regarding the results, limitations of the study and further testing platform development for more pertinence.

Reviewer 3 Report

The manuscript discusses an interesting framework for remote usability assessment as well as the evaluation of the public tax portal in Slovenia, by two different groups of people, namely adults and elderly. Overall, the provided methodology is rigorous with sound results, however the authors should take into consideration the following points that will improve the comprehension of the readers:

  • Regarding the background section, the authors should consider reducing the sub sections that refer to well-known methods and standards such as UCD, heuristic evaluation proposed by Nielsen et.al, etc., in order to avoid tiring readers and also gain more space so as to provide further details regarding the proposed FCD approach (see also point below).
  • The background information regarding the FCD presented in the manuscript lacks concrete evidence in respect of how it differentiates from UCD, what are the key features and objectives and why it is considered a better approach in comparison to widely established similar methodologies.
  • Regarding the discussion related to the testing platform, more under the hood details are expected to be provided in the newer version illustrating what are the prerequisites and the effort that a third party needs to invest so as to setup a new experiment. Furthermore, the authors should suggest how they are planning to support qualitative cues provided during a physical usability evaluation, such as capturing user’s verbal feedback while they are executing a task, the level of their attention, fatigue, etc.
  • Regarding the experiment description, I suggest to the authors to provide more details regarding the design phase of the alternative interfaces, highlighting how they employed the FCD process and entailing what are the different aspects that were considered for the different end-user groups. It should be also clarified whether the authors involved representative end-users in the design process. Additionally, it would be beneficial for the readers’ comprehension if the authors provided concrete examples that illustrate the different aspects that have been taken into consideration for designing the interfaces for the different groups.
  • Further to the above-mentioned comment, the authors should provide further details regarding their finding that the time-on-task was improved in the new interfaces, mainly because of the re-design of the web forms.
  • The authors should argue on their selection to use different web content accessibility levels for the two end-user groups, and how this choice is reflected in the experiment results.
  • Finally a general comment is that although the authors emphasize the FCD process which was followed, the end-user sample that has been selected seems to not have family relations. Thus, the presented work is more an assessment for improving the UX and effectiveness of web UIs for the elderly rather than an FCD flavoured evaluation.

Author Response

Dear Reviewer,

thank you for reviewing the manuscript. We greatly appreciate your comments and recommendations that help us improve the article. We have revisited the manuscript accordingly and made structural and content related changes. As there is a considerable amount of updates in the manuscript, while using “track changes” document mode we recommend to use the “Simple Markup” option for optimal reading experience.

The following is our point-by-point responses. We hope that you find our responses satisfactory.

---

The manuscript discusses an interesting framework for remote usability assessment as well as the evaluation of the public tax portal in Slovenia, by two different groups of people, namely adults and elderly. Overall, the provided methodology is rigorous with sound results, however the authors should take into consideration the following points that will improve the comprehension of the readers:

Point 1: Regarding the background section, the authors should consider reducing the sub sections that refer to well-known methods and standards such as UCD, heuristic evaluation proposed by Nielsen et.al, etc., in order to avoid tiring readers and also gain more space so as to provide further details regarding the proposed FCD approach (see also point below).

Response 1: The mention of Nielson’s methods remains to illustrate the diversity of approaches in the UX field, and authors included additional descriptions on standard UX measurement tools used within this experiment to amend other Reviewers. Further details regarding FCD were added, see next point (line 330).

Point 2: The background information regarding the FCD presented in the manuscript lacks concrete evidence in respect of how it differentiates from UCD, what are the key features and objectives and why it is considered a better approach in comparison to widely established similar methodologies.

Response 2: In the Background section the authors included additional characterization related to FCD concept (line 330). Throughout the manuscript the authors try to appeal to reader that FCD approach is a concept organized around main theme, the established UCD methodology, and tries to improve certain aspects of the latter.

Point 3: Regarding the discussion related to the testing platform, more under the hood details are expected to be provided in the newer version illustrating what are the prerequisites and the effort that a third party needs to invest so as to setup a new experiment.

Response 3: The new subsection 5.3 Testing platform and future development under 5. Discussion section now includes the testing platform transferability potential, basic technology requirements and time related investments for third parties and an invitation to contact the authors for code and instructions (line 1253).

Point 4: Furthermore, the authors should suggest how they are planning to support qualitative cues provided during a physical usability evaluation, such as capturing user’s verbal feedback while they are executing a task, the level of their attention, fatigue, etc.

Response 4: The additional subsection on 5.2 Limitations of the study within the 5. Discussion section deliberates on particular technical choices (e.g. eye tracking with web camera, and monitoring participants via web camera) in experiment design (line 1152) that have been tested in laboratory setting environment, and how to leverage that approach that allows for more qualitative cues with online remote testing, that also has its benefits (larger sample size, mitigation of low participants’ digital competences, no moderator present etc.).

Point 5: Regarding the experiment description, I suggest to the authors to provide more details regarding the design phase of the alternative interfaces, highlighting how they employed the FCD process and entailing what are the different aspects that were considered for the different end-user groups. It should be also clarified whether the authors involved representative end-users in the design process. Additionally, it would be beneficial for the readers’ comprehension if the authors provided concrete examples that illustrate the different aspects that have been taken into consideration for designing the interfaces for the different groups.

Response 5: Section 3.2.2. Testing interface definition has been substantially rewritten to connect FCD concepts with WCAG 2.1 and interface conception. The accessibility improvements for both generations in the form of WCAG 2.1 AA (adults) and AAA (elderly) success criteria were central to amend the FCD concept in part concerned with different levels of digital competences among generations (line 534). The motivation to implement two distinct FCD-based interfaces has indeed derived from responses of a small focus group, the comment on that aspect has been added to 3.2.2. Testing interface definition section (line: 557).

Point 6: Further to the above-mentioned comment, the authors should provide further details regarding their finding that the time-on-task was improved in the new interfaces, mainly because of the re-design of the web forms. Further to the above-mentioned comment, the authors should provide further details

Response 6: Thank you for your important remark. The authors have included the overall time-on-task improvement comment in the 5.1 Implications of the results section (line 1124), and tried to explain that the enhancements were exclusively on perceivable and understandable principles, without any service delivery changes that would provide shortcuts or bypasses not available in the original eDavki interface as described in 3.2.2. Testing interface definition section (line 600). Additionally, the task 3, tax form filling, has substantially shorter time on task metric, there were no changes in terms of content, merely color palette, sizing and spacing of the text, content layout was equally linear yet content sections were presented in panels in accordance to Table 1 (line 595) in 3.2.2. Testing interface definition section.

Point 7: The authors should argue on their selection to use different web content accessibility levels for the two end-user groups, and how this choice is reflected in the experiment results.

Response 7: The conception stage of interfaces was additionally described in 3.2.2. Testing interface definition (line 534). The choice for using two WCAG 2.1 success criteria levels for each of the generations’ interface was fixed with generational membership within FCD-based interface (Interface 2), so no direct comparison between the AA and AAA levels (within same user group) was presented, rather it is separately addressed as FCD interface for adults (Generation 1) and elderly (Generation 2), thus including generation as an additional factor, later throughout the article.

Point 8: Finally a general comment is that although the authors emphasize the FCD process which was followed, the end-user sample that has been selected seems to not have family relations. Thus, the presented work is more an assessment for improving the UX and effectiveness of web UIs for the elderly rather than an FCD flavoured evaluation.

Response 8: Unfortunately it was not possible to implement all important components of FCD concept, such as group involvement in joint processes supported by the use of digital solutions and concern for multigenerational communities (family relations) - the researchers were able to include all these FCD tenets (line 331) in previous research work where the common MyFamily application has been developed, yet, in respect to generations participating (no youth) and established tax portal scenarios the FCD approach was only partially included, in the part concerned with different levels of digital competences among generations. To support the general digitalization efforts the researchers perceive that the need to improve access and usability of digital services for all generations (and competence levels) is very important. Concerns related to you point were also expressed in the 5. Discussion section in relation to Limitations of the study (1151).

Round 2

Reviewer 1 Report

Dear authors

Thank you for your response. I think that you have responded adequately to the majority of my comments.

However, it is not still clear to me:

  1. how did you incorporate the family concept into your sample? Please explain more thoroughly.  
  2. In table 3 how have you calculated the performance metrics of Interface 2 as you have two interfaces FCD adult FCD elderly. 

Author Response

Dear Reviewer,

thank you once again for reviewing the manuscript.

We appreciate your recommendations that helped to improve the article immensely. We have revised the manuscript and made content related changes according to your comments.

Again, there is a considerable amount of updates in the manuscript, while using “track changes” document mode we recommend to use the “Simple Markup” option for optimal reading experience.

Please find below the point-by-point responses. We hope that responses and updates to the manuscript are satisfactory.

Best regards, The Authors

--

Dear authors.

Thank you for your response. I think that you have responded adequately to the majority of my comments. However, it is not still clear to me:

Point 1: how did you incorporate the family concept into your sample? Please explain more thoroughly. 

Response 1:  In the background section the authors included additional characterization related to FCD concept (line 331). Throughout the manuscript the authors try to appeal to reader that FCD approach is a concept organized around main theme, the established UCD methodology, and tries to improve certain aspects of the latter. Unfortunately, it was not possible to implement all important components of FCD concept, such as group involvement in joint processes supported by the use of digital solutions and concern for multigenerational communities (family relations) in this experiment. The general improvement would be to improve access and usability of digital services for all generations (and competence levels).

Samples of adult (aged 18-64) and elderly (aged 65+) population selection was based on:

  • Developmental tasks based separation (line 908) on adult and elderly population has been used in relation to life events and actions that are addressed in interaction with tax authorities thru various tax forms. Some of these forms are applicable only for working adults (e.g. social security contributions) while others are applicable mostly for pensioners.
  • The researchers were able to include all FCD principles (line 331) in previous research work where the common MyFamily application has been developed and testing has been conducted with same population strata, so it made sense to researchers to use same samples’ definitions and same performance and user-evaluation metrics for future comparability purposes.
  • In respect to generations participating (no youth) in established tax portal scenarios (intended for adults) the FCD approach was only partially included, in the part concerned with different levels of digital competences among generations.

Concerns related to this important point were also expressed in the 5. Discussion section in relation to Limitations of the study (1194).

Point 2: In table 3 how have you calculated the performance metrics of Interface 2 as you have two interfaces FCD adult FCD elderly.

Response 2:  The performance metrics, for example in Table 3, were calculated for total average of all tasks within an interface (original or FCD), while horizontal bar graph, e.g. Figure 8 for time on task, shows a distribution of averages per 3 tasks together. Some clarification on this has been added to the manuscript (line 1024 and 1067). Moreover, each single task statistics and tests were presented in the Supplement 1, for example in tables S13 (success) and S14 (time on task), while the overall, cumulative (all tasks together) performance metrics per interface or generation have been included in the manuscript.

The FCD interface data was treated as a single interface. This interface was adaptive, as one version has been seen only by adults and other version has been displayed only for elderly.

Additionally, the authors have included the overall time-on-task improvement comment in the 5.1 Implications of the results section (line 1139), and tried to explain that the enhancements were exclusively on perceivable and understandable principles, without any service delivery changes that would provide shortcuts or bypasses not available in the original eDavki interface. The task 3, tax form filling, had shorter time on task metric, however there were no changes in terms of content, merely color palette, sizing and spacing of the text, content layout was equally linear yet content sections were presented in panels in accordance to Table 1 (line 596) in 3.2.2. Testing interface definition section.

Also, please note there was a substantial redesign of statistical method for finding differences among groups, which is described in 4.1. Method section (line 820). Most result were similar to previous instance and did not have decisive impact on the article’s narrative, however this has improved the article’s methodological soundness.

Reviewer 2 Report

I am still not convinced that the authors should have employed the parametric t-test before they have tested all the assumptions that needs to be met. In that respect, the authors are asked to provide evidence that the data they collected comply with all relevant assumptions, especially when the normality of data is considered (with e.g. results of Shapiro-Wilk test). It is true that in the literature one can find evidence that t-test can be applied when sample size is large enough which is not the case in the research presented in this paper. The authors should also check grammar, especially in the newly written parts of the texts because it is full of grammatical errors and use of informal terms. 

Author Response

Dear Reviewer,

thank you once again for reviewing the manuscript.

We appreciate your recommendations that helped to improve the article immensely. We have revised the manuscript and made content related changes according to your comments.

Again, there is a considerable amount of updates in the manuscript, while using “track changes” document mode we recommend to use the “Simple Markup” option for optimal reading experience.

Please find below the point-by-point responses. We hope that responses and updates to the manuscript are satisfactory.

Best regards, The Authors

--

Point 1: I am still not convinced that the authors should have employed the parametric t-test before they have tested all the assumptions that needs to be met. In that respect, the authors are asked to provide evidence that the data they collected comply with all relevant assumptions, especially when the normality of data is considered (with e.g. results of Shapiro-Wilk test). It is true that in the literature one can find evidence that t-test can be applied when sample size is large enough which is not the case in the research presented in this paper.

Response 1:  Thank you for insisting on the importance of adequate quantitative methods selection. The researchers have revisited the statistical methods and implemented Shapiro-Wilk test for normal distribution on all samples compared in the article (please see Tables S11 and S12 in the Supplement 1). Shapiro-Wilk tests did not confirm normal distribution on majority of samples, thus a non-parametric equivalent of the two sample t-test, The Mann-Whitney U test (line 828) has been selected to compare differences among samples. The results on all pairs of samples (cumulatively 84 times, see Supplement 1) have been recalculated. Conforming to null hypothesis framework, results are presented in tables and findings throughout the manuscript. Most result were similar to previous instance and did not have decisive impact on the article’s narrative, however this has improved the article’s methodological soundness.

Moreover, your input contributed to decision on future implementation of formal normality tests in the testing platform’s backend analysis tool to help select appropriate statistical method in seeking statistically significant differences.

Point 2: The authors should also check grammar, especially in the newly written parts of the texts because it is full of grammatical errors and use of informal terms. 

Response 2: The grammar has been checked by professional proofreader and the authors revised newly written parts and hopefully made some improvements in the vocabulary.